# Inner Product-based Neural Network Similarity

**Wei Chen**[†], **Zichen Miao**[†], **Qiang Qiu**
Department of ECE
Purdue University
{chen2732, miaoz, qqiu}@purdue.edu

## Abstract

Analyzing representational similarity among neural networks (NNs) is essential for interpreting or transferring deep models. In application scenarios where numerous NN models are learned, it becomes crucial to assess model similarities in computationally efficient ways. In this paper, we propose a new paradigm for reducing NN representational similarity to filter subspace distance. Specifically, when convolutional filters are decomposed as a linear combination of a set of filter subspace elements, denoted as *filter atoms*, and have those decomposed atom coefficients shared across networks, NN representational similarity can be significantly simplified as calculating the cosine distance among respective filter atoms, to achieve *millions of times* computation reduction over popular probing-based methods. We provide both theoretical and empirical evidence that such simplified filter subspace-based similarity preserves a strong linear correlation with other popular probing-based metrics, while being significantly more efficient to obtain and robust to probing data. We further validate the effectiveness of the proposed method in various application scenarios where numerous models exist, such as federated and continual learning as well as analyzing training dynamics. We hope our findings can help further explorations of real-time large-scale representational similarity analysis in neural networks.

## 1 Introduction

Deep neural networks (NNs) have shown unprecedented performance in a large variety of tasks [24, 46]. In many scenarios, numerous models are learned and their relations can be beneficial to exploit. For example, as illustrated in Figure 1(a), to aggregate knowledge across space, federated learning (FL) trains models over a large number of clients while keeping data localized. To preserve knowledge across time while learning new ones, continual learning (CL) can be addressed by training a large group of models, one for each timestep. Finding the relations among models is the cornerstone to boosting performance in these scenarios, such as improving personalization for FL [53] or providing knowledge retrieval for CL [40]. Considering the large number of NNs potentially allowed in those scenarios, e.g., to model growing spatial/temporal coverage, it becomes crucial to have a highly computationally efficient way to assess NN model similarity.

We are inspired by one recent state-of-the-art CL framework in [35], where each convolutional filter is represented as a linear combination of a set of filter subspace elements, denoted as *filter atoms*. It is easy to notice that each convolutional layer now becomes two convolutional layers, a filter atom layer followed by an atom coefficient layer with $1 \times 1$ filters. Then, motivated by the literature on task subspace modeling [10, 25, 33, 45, 64] that tasks can be modeled as a set of latent basis tasks and their linear combinations, a group of tasks are sequentially modeled using NNs by learning for each task a different set of filter atoms, while sharing common atom coefficients across

---

[†]Equal contribution.

37th Conference on Neural Information Processing Systems (NeurIPS 2023).

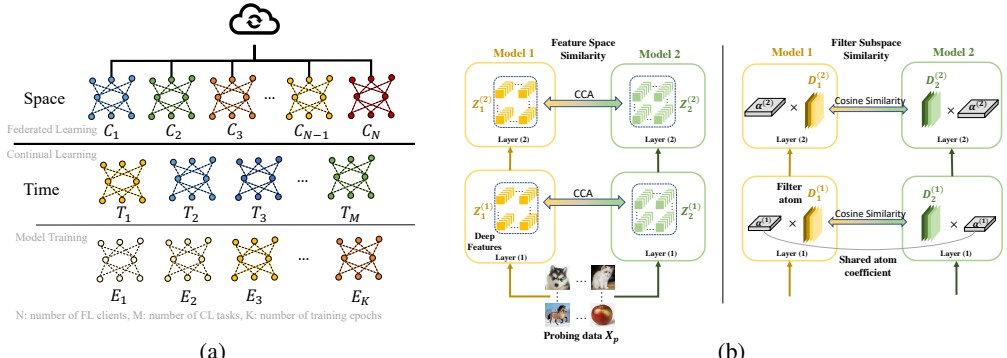

(a)                                                                 (b)

Figure 1: (a) The illustration of scenarios where numerous models exist, such as federated learning (FL), continual learning (CL), and model training process. The relations among models are usually critical, and the computational cost to assess the model relation can be a major bottleneck. (b) Comparison between our method and probing-based methods. (left) Feature space similarity metrics, e.g., CCA, rely on probing data, and calculate the correlation between large groups of features generated by the forward pass of probing data through NNs. (right) In comparison, our filter subspace-based method decomposes convolutional filters $\mathbf{W}$ as *filter atoms* $\mathbf{D}$ (filter subspace elements) and *atom coefficients* $\boldsymbol{\alpha}$, $\mathbf{W} = \boldsymbol{\alpha} \times \mathbf{D}$, and only calculates the filter subspace similarity between a small portion of parameters, *i.e.*, filter atoms, which is independent from probing data and computation efficient. The proposed filter subspace-based method can achieve *millions of times* computation reduction than popular probing-based methods.

tasks. [35] has in detail analyzed and validated this framework in the CL context. This learning framework with individually modeled filter subspaces but shared rules of linear combinations is generally applicable to many multi-model scenarios, especially CL and FL settings, where numerous NNs with the same architecture are learned. For example, FL learns a single global model to fit the data on all clients, and the majority of aggregating methods in FL require NNs to maintain the same network structure [28, 29, 34].

In the above setting, it is easy to observe that the representation variations across different NNs now become dominated by respective filter atoms. Thus, [35] adopts in experiments filter subspace distance to assess task relevancy, however, without formal justification. In this paper, we formally explore NN representational similarity using filter subspace distance, with detailed theoretical and empirical justifications. We first simplify the filter subspace distance to the cosine distance of two sets of filter atoms, to eliminate the computation of singular value decomposition in calculating principal angles. Then, we show both theoretically and empirically that the obtained filter subspace similarity preserves a strong linear correlation with other popular probing-based similarity measures such as CCA [43], which require external probing data as input stimuli. Our representational similarity is also immune to inappropriate choices of probing data, while probing-based metrics can be perturbed drastically.

Previous works [37, 43] measure representational similarity directly relying on deep representations revealed by input data. These approaches introduce heavy computation from both the forward pass of numerous probing data and the calculation of high-dimensional covariance matrices. As these similarity metrics are probing-dependent, their quality can potentially deteriorate when probing data are inappropriately chosen, scarce or unavailable. Such properties make the popular probing-based approaches less appropriate for our target scenarios where a large number of NN models are present.

The proposed filter subspace similarity shows extreme efficiency in both memory and computation. Since our similarity computation does not involve network forward pass, no GPU memory access is required, whereas other probing-based measures consume the same amount of GPU memory as regular inference. On the other hand, the proposed method involves only inner product calculations on filter atoms, which takes negligible time for similarity evaluation. The evaluation time of probing-based measures includes the time of both the forward pass of probing data and the calculation of high-dimensional covariance matrices. We report later the dramatically improved evaluation time of the proposed method against other popular probing-based methods, e.g., CKA [21]. These unique

properties make our method highly desirable for exploring NN similarity under scenarios with a large number of NN models.

We further validate our filter subspace similarity for knowledge transfer with various CL and FL tasks, as sample examples to exploit NN model relations. In both settings, we fix the atom coefficients, learn the filter atoms for each task, and finally conduct knowledge transfer among tasks by recalling the most similar models for the ensemble. Compared with probing-based similarity metrics, the proposed measure achieves competitive performance with *millions of times* reduction in the computational cost.

We summarize our contributions as follows,

- We formally explore NN representational similarity measure using filter subspace distance.
- We show both theoretically and empirically that the proposed filter subspace-based measure preserves a strong linear correlation with other popular probing-based measures, while being significantly more robust and efficient in both memory and computation.
- We demonstrate the effectiveness of the proposed similarity measure using several simple examples, such as federated and continual learning as well as analyzing training dynamics.

## 2    Methodology

In this section, we first review probing-based representational similarities and show their limitations. Then, we provide a filter subspace formulation for NNs, and propose a NN similarity metric based on a simplified filter subspace distance. We further demonstrate that under certain assumptions, the proposed measure shows a strong linear relationship with popular probing-based measures, while exhibiting dramatic improvement in computational efficiency and data robustness. These unique characteristics of the proposed measure can potentially enable real-time large-scale NN similarity assessment, e.g., helping fast knowledge retrieval across a large number of NN models.

### 2.1    Revisiting Representational Similarity in Feature Space

Intuitively, the NN representational similarity can be directly assessed via features generated from different neural networks. As shown in Figure 1(b), it usually includes three steps to evaluate probing-based representational similarity between two NNs $\mathcal{F}_u$ and $\mathcal{F}_v$: (1) Collect an appropriate and sufficient amount of external probing data $\mathbf{X}_p \in \mathbb{R}^{n \times c' \times h' \times w'}$ that can represent the whole data distribution. (2) Generate the feature $\mathbf{Z}_u$ and $\mathbf{Z}_v$ ($\mathbf{Z}_u, \mathbf{Z}_v \in \mathbb{R}^{n \times c \times h \times w}$) by the forward pass of probing data through different neural networks, $\mathbf{Z}_u = \mathcal{F}_u(\mathbf{X}_p, \theta_u)$ and $\mathbf{Z}_v = \mathcal{F}_v(\mathbf{X}_p, \theta_v)$, where $\theta_u, \theta_v$ denote parameters of two NNs. (3) Choose a probing-based metric to assess the model similarity. Several popular probing-based methods can be adopted in step (3), and we will give a brief introduction below.

**CCA.**    [43] proposes to analyze the NN representational similarity by conducting canonical correlation analysis on $\mathbf{Z}_u, \mathbf{Z}_v$, which is a recursive process of finding projection directions for two matrices that their correlation is maximized. Specifically, let $Q_u, Q_v$ denote the orthonormal bases of $\mathbf{Z}_u, \mathbf{Z}_v$, the CCA can be denoted as,

$$\mathcal{S}_{CCA}(\mathcal{F}_u, \mathcal{F}_v) = \sqrt{\frac{1}{c} \sum_{l=1}^{c} \sigma_l^2}, \tag{1}$$

where $\sigma_l$ denotes the $l$-th eigenvalue of $\Lambda_{u,v} = Q_u^\mathsf{T} Q_v$.

**CKA.**    [21] proposes another way to assess the NN similarity based on Centered Kernel Alignment (CKA). Let $K_u = \mathbf{Z}_u \mathbf{Z}_u^\mathsf{T}, K_v = \mathbf{Z}_v \mathbf{Z}_v^\mathsf{T}$ denote the Gram matrices of two feature space, the CKA is computed by,

$$\mathcal{S}_{CKA}(\mathcal{F}_u, \mathcal{F}_v) = \frac{\mathrm{HSIC}(K_u, K_v)}{\sqrt{\mathrm{HSIC}(K_u, K_v)\, \mathrm{HSIC}(K_u, K_v)}}, \tag{2}$$

where HSIC is the Hilbert-Schmidt Independence Criterion [11].

However, in addition to the forward pass, all the aforementioned approaches further introduce significant computational costs while performing evaluation in the representation space. Nevertheless,

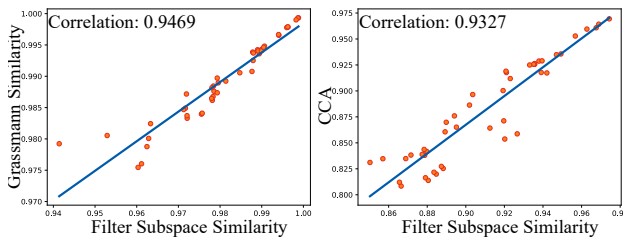

| | Correlation |
| --- | --- |
| CCA [43] | 0.9327 |
| CKA [21] | 0.9550 |
| Grassmann Distance [35] | 0.9469 |

Figure 2: (a) Correlation between Grassmann similarity and filter subspace similarity; (b) Correlation between CCA and filter subspace similarity. (Table) Correlation between filter subspace similarity and other approaches.

their qualities rely heavily on the mindful choice of probing data $\mathbf{X}_p$, which undermines their robustness.

## 2.2 Representational Similarity in Filter Subspace

**Filter subspace.** As in [41], the convolutional filter $\mathbf{W} \in \mathbb{R}^{c' \times c \times k \times k}$ ($c'$ and $c$ are the number of input and output channels, $k$ is the kernel size) can be decomposed over $m$ *filter atoms* (filter subspace elements) $\mathbf{D}[i] \in \mathbb{R}^{k \times k} (i = 1, ..., m)$, linearly combined by *atom coefficients* $\boldsymbol{\alpha} \in \mathbb{R}^{m \times c' \times c}$ as $\mathbf{W} = \boldsymbol{\alpha} \times \mathbf{D}$. Note that each convolutional layer now becomes two convolutional layers, a filter atom layer followed by an atom coefficient layer with $1 \times 1$ filters. The filter subspace is then expressed as $\mathcal{V} = \mathrm{Span}\{\mathbf{D}[1], ..., \mathbf{D}[m]\}$. With this formulation, we consider a paradigm where filter subspaces are model-specific, and subspace linear combination rules, *i.e.*, atom coefficients, are shared across different networks. The intuition and detailed validation of this learning paradigm can be found in [35], where state-of-the-art performance in the continual learning context is reported.

In this setting, we dive deep into the relationship between filter subspaces and representations. For simplicity, let $c = c' = 1$, and the argument extends. Given an input image $\mathbf{X}(b)$ ($b \in \mathcal{B}, \mathcal{B} \subset \mathbb{Z}^2$), define the local input norm $||\mathbf{X}||_{F, N_b} := (\sum_{b' \in N_b} \mathbf{X}(b - b')^2)^{1/2}$ and the convolution $\langle \mathbf{X}, w \rangle_{N_b} := \sum_{b' \in N_b} \mathbf{X}(b - b')w(b')$, where $N_b \subset \mathcal{B}$ is a local Euclidean grid centered at $b$. Then the decomposed convolution can be written as $\mathbf{Z}(b) = \sum_{i=1}^{m} \boldsymbol{\alpha}_i \langle \mathbf{X}, \mathbf{D}_i \rangle_{N_b}$, where $\mathbf{D}[i]$ denotes the $i$-th atom, $\boldsymbol{\alpha}_i$ is the corresponded $i$-th coefficient.

**Proposition 2.1.** *Suppose $\mathbf{D}_u$ and $\mathbf{D}_v$ are two different sets of filter atoms for a convolutional layer with the common atom coefficients $\boldsymbol{\alpha}$, we can upper bound the changes in the corresponding features $\mathbf{Z}_u, \mathbf{Z}_v$ with atom changes,*

$$||\mathbf{Z}_u - \mathbf{Z}_v||_F \leq (||\boldsymbol{\alpha}||_F \lambda) \sqrt{|\mathcal{B}|} \cdot ||\mathbf{D}_u - \mathbf{D}_v||_F, \quad with \ \lambda = \sup_{b \in \mathcal{B}} ||\mathbf{X}||_{F, N_b}. \tag{3}$$

The proof is provided in Appendix A.1. We further empirically validate this relationship in Section A.3.

**Filter subspace similarity** The above theorem suggests the possibility to measure the representational similarity of two NNs by simply measuring the distance of their filter subspaces. As proposed in [35], the representational similarity of two NNs with different filter subspaces $\mathcal{V}_u, \mathcal{V}_v$ can be assessed by the similarity based on Grassmann distance between $\mathcal{V}_u, \mathcal{V}_v$ as,

$$\mathcal{S}_{Gras}(\mathcal{F}_u, \mathcal{F}_v) = d(\mathcal{V}_u, \mathcal{V}_v) = \frac{1}{m} \sum_i \cos\theta_i, \tag{4}$$

where $\theta_i$ is the $i$-th principal angle between $\mathcal{V}_u$ and $\mathcal{V}_v$.

However, the above metric requires costly singular value decomposition. Note that filter atoms in different NNs are intrinsically aligned under shared atom coefficients, which allows us to approximate the filter subspace similarity using the cosine similarity of the corresponding filter atoms. To this end, as shown in Figure 1(b), we propose a significantly simplified representational similarity measure with filter atom similarity.

**Definition 2.2.** Suppose two convolution neural networks $\mathcal{F}_u, \mathcal{F}_v$ share atom coefficients layer-wise, and their model-specific filter atoms are $\mathbf{D}_u, \mathbf{D}_v$, then the filter subspace representational similarity is simplified as,

$$\mathcal{S}_{Atom}(\mathcal{F}_u, \mathcal{F}_v) = \mathbf{cos}(\mathbf{D}_u, \mathbf{D}_v) = \frac{< vec(\mathbf{D}_u), vec(\mathbf{D}_v) >}{||vec(\mathbf{D}_u)|| \cdot ||vec(\mathbf{D}_v)||}. \tag{5}$$

The above definition is a layer-wise similarity, allowing us to compare the similarity of different networks per layer, and we simply average layer-wise similarities for the network-wise similarity.

*Remark* 2.3. The filter subspace similarity measure becomes a proper metric after taking the arccosine, *i.e.*, $\arccos(\mathcal{S}_{Atom}(\mathcal{F}_u, \mathcal{F}_v))$ is a proper metric.

We further show that $\mathcal{S}_{Atom}$ and $\mathcal{S}_{Gras}$ are equivalent under certain assumption.

**Proposition 2.4.** *Assume* $\mathbf{D}_u, \mathbf{D}_v \in \mathbb{R}^{k^2 \times m}$ *are orthogonal matrices, then* $\mathcal{S}_{Gras} = \mathcal{S}_{Atom}$.

The proof is provided in Appendix A.1. We empirically show in Figure 2(a) that the above simplified filter subspace similarity has still a strong linear correlation with the Grassmann subspace similarity even without imposing the above orthogonality over atoms.

Note that our filter subspace similarity measure only involves linear operations of vectorized atoms of around hundreds of dimensions, which requires negligible computation. Additionally, the proposed method depends solely on models themselves and eliminates the reliance on external probing data, equipping our similarity with robustness to inappropriate choice of probing data.

## 2.3 Algorithm Complexity Analysis

Here, we provide a detailed comparison of computation complexity between the proposed filter subspace similarity and probing-based similarities. Consider one convolutional layer with filter $\mathbf{W} \in \mathbb{R}^{c' \times c \times k \times k}$ ($\mathbf{W} = \boldsymbol{\alpha} \times \mathbf{D}, \mathbf{D} \in \mathbb{R}^{m \times k \times k}$) which transforms the input $\mathbf{X}_p \in \mathbb{R}^{n \times c' \times h' \times w'}$ to output $\mathbf{Z} \in \mathbb{R}^{n \times c \times h \times w}$. The complexity of our method is dominated by inner product of two tiny filter atoms, $\mathcal{O}(m \cdot k^2)$, *e.g.*, $m = 9, k = 3$ in a typical setting.

In contrast, probing-based similarity measure first forward feeds $n$ probing samples with a complexity of $\mathcal{O}(n \cdot h'w' \cdot k^2 \cdot cc')$, then calculates covariance matrix with the complexity of $\mathcal{O}(n^2 \cdot hw \cdot c)$. In total, the time complexity of CCA is $\mathcal{O}(n \cdot h'w' \cdot k^2 \cdot cc' + n^2 \cdot hw \cdot c)$. Our method is at least $\frac{n \cdot h'w' \cdot k^2 \cdot cc' + n^2 \cdot hw \cdot c}{m \cdot k^2}$ times more efficient than probing-based similarity measures. As $h \gg k$, $cc' \gg m$, the computational cost of our method is negligible. For example, with 10k probing datapoints, the CCA calculation requires $1.14 \times 10^7$ times more FLOPs than the proposed method.

## 2.4 Relationship with Probing-based Similarities

The proposed filter subspace similarity not only shows extreme efficiency but also exhibits a strong linear relationship with other popular probing-based similarities. Here, we analyze the proposed filter subspace similarity $\mathcal{S}_{Atom}$ with CCA, $\mathcal{S}_{CCA}$ [43]. Suppose forward passes of decomposed convolutional layer for $\mathcal{F}_u$ and $\mathcal{F}_v$ are $\mathbf{Z}_u = \boldsymbol{\alpha}\mathbf{X}_p\mathbf{D}_u, \mathbf{Z}_v = \boldsymbol{\alpha}\mathbf{X}_p\mathbf{D}_v$, respectively. [†] To start with, we show that the $\mathcal{S}_{CCA}$ is upper bounded by the proposed $\mathcal{S}_{Atom}$.

**Theorem 2.5.** *Let* $\mathcal{T} = \mathbf{Tr}(\mathbf{X}_p^\intercal \boldsymbol{\alpha}^\intercal \boldsymbol{\alpha}\mathbf{X}_p), \mathcal{C} = \sigma_{min}(\mathbf{X}_p^\intercal \boldsymbol{\alpha}^\intercal \boldsymbol{\alpha}\mathbf{X}_p)$. *Assume* $\mathcal{K}(\mathbf{Z}_u^\intercal \mathbf{Z}_u), \mathcal{K}(\mathbf{Z}_v^\intercal \mathbf{Z}_v) \leq \gamma$. *Then* $\mathcal{S}_{CCA}(\mathcal{F}_u, \mathcal{F}_v)$ *is upper bounded by* $\mathcal{S}_{Atom}(\mathcal{F}_u, \mathcal{F}_v)$,

$$\frac{\mathcal{C}}{\gamma c^{\frac{3}{2}} \mathcal{T}} \cdot \mathcal{S}_{CCA}(\mathcal{F}_u, \mathcal{F}_v) \leq \mathcal{S}_{Atom}(\mathcal{F}_u, \mathcal{F}_v), \tag{6}$$

where $\mathbf{Tr}(\cdot)$ denotes trace of a matrix, $\sigma_{min}$ indicates the minimum eigenvalue, $\mathcal{K}(A)$ denotes the condition number of matrix $A$. We provide the proof in Appendix A.1.

---

[†]Specifically, the formulation writes as $\mathbf{Z}_u = \boldsymbol{\alpha}\mathbf{X}_p \star \mathbf{D}_u$, where $\star$ is the convolutional operation. By converting convolutional kernel $\mathbf{D}_u$ into a Toeplitz matrix, we can replace the convolution operation $\mathbf{X}_p \star \mathbf{D}_u$ with matrix multiplication $\mathbf{X}_p\mathbf{D}_u$. We also modify $\alpha$ by $I_{hw} \bigotimes \alpha$, where $\bigotimes$ is Kronecker product, to enable the matrix multiplication $\boldsymbol{\alpha}\mathbf{X}_p\mathbf{D}_u$.

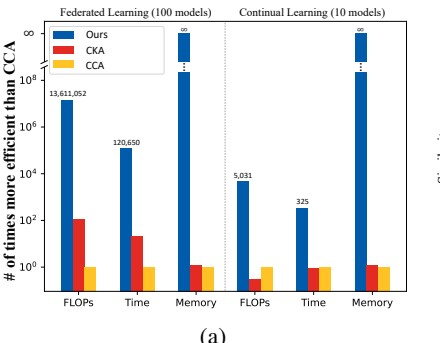
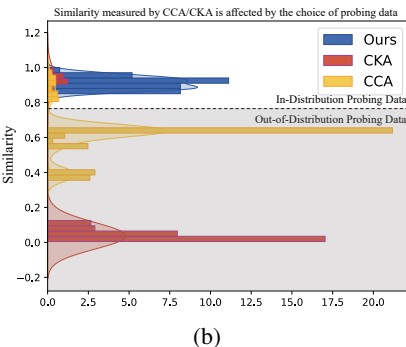

(a)                                 (b)

Figure 3: (a) The ratio of the computational cost savings of our filter subspace similarity over probing-based similarities. (b) The performance of probing-based similarities can be compromised by poorly selected probing data. For models trained on CIFAR-100, they have high CCA and CKA similarities with probing from CIFAR-100 but low similarities with probing from other datasets. In contrast, our filter subspace similarity does not rely on probing data and shows a high similarity between the networks, aligning with our expectations.

Since $\mathcal{S}_{CCA}$ is probing-dependent, the calculated value varies depending on the choice of probing data, and the value range shows bounded by our filter subspace similarity, as in the theorem above.

With additional assumptions imposed, we can further show a near-linear relationship between CCA and our filter subspace similarity.

**Assumption 2.6.** Suppose the diagonal elements of $\mathbf{Z}_u^\intercal \mathbf{Z}_u$, $\mathbf{Z}_u^\intercal \mathbf{Z}_v$ and $\mathbf{Z}_v^\intercal \mathbf{Z}_v$ are larger than non-diagonal element, *i.e.*, $(\mathbf{Z}_u^\intercal \mathbf{Z}_u)_{ii} \gg (\mathbf{Z}_u^\intercal \mathbf{Z}_u)_{ij}$.

The Assumption 2.6 suggests different channels of feature $\mathbf{Z}$ have a low correlation. Reducing channel-wise dependencies has been studied in [63] and has been shown to benefit model stability. We provide the empirical verification of the assumption in Appendix A.3.

**Theorem 2.7.** *If Assumption 2.6 holds, $\mathcal{S}_{CCA}(\mathcal{F}_u, \mathcal{F}_v)$ is approximately linear to filter subspace similarity,*

$$\frac{\sqrt{c}}{\gamma_1 \gamma_2 \gamma_3} \cdot \mathcal{S}_{CCA}(\mathcal{F}_u, \mathcal{F}_v) = \mathcal{S}_{Atom}(\mathcal{F}_u, \mathcal{F}_v), \tag{7}$$

where $\gamma_1$, $\gamma_2$ and $\gamma_3$ contain higher order of features, which can be found in detail with the proof in Appendix A.1. Specifically, we have $\gamma_2 = \sqrt{1 - \frac{\Delta}{\gamma_1^2 \gamma_3^2} \frac{1}{cos^2(\mathbf{D}_u, \mathbf{D}_v)}}$, and since $\Delta$ are small, with Taylor expansion, $\gamma_2 \approx 1 - \frac{1}{2} \frac{\Delta}{\gamma_1^2 \gamma_3^2} \frac{1}{cos^2(\mathbf{D}_u, \mathbf{D}_v)}$. The term $\frac{1}{cos^2(\mathbf{D}_u, \mathbf{D}_v)}$ causes non-linearity in the relation between CCA and filter subspace similarity.

As in Figure 2, we empirically observe the linear correlation between CCA and filter subspace similarity, which agrees with our theoretical findings. In addition, we find that the proposed similarity also shows a strong correlation with CKA.

## 3 Experiments

In this section, we first validate our theorems with several validation experiments and then demonstrate simple example applications of the proposed filter subspace similarity in efficiently analyzing training dynamics as well as in federated and continual learning scenarios.

### 3.1 Validation Experiments

We conduct empirical validation to confirm the near-linear relationship between filter subspace similarity and probing-based similarity and explored the limitations of probing-based similarities.

**Correlation of CCA and filter subspace similarity.** The empirical verification of the correlation between CCA and filter subspace similarity is presented in Figure 2. In this experiment, 10 tasks are

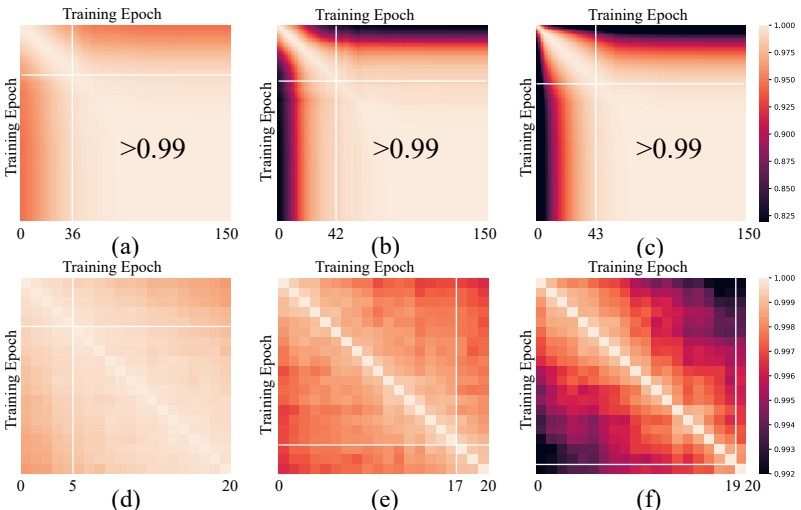

Figure 4: Layer-wise similarity matrices that show relations of model parameters of different training time points. (a)(b)(c) are the 1st, 3rd and 5th convolutional layer of AlexNet trained on CIFAR-100. (d)(e)(f) are the 1st, 4th and 8th convolutional layer of VGG11 trained on ImageNet. We mark the epoch when the parameter reaches 0.99/ 0.999 similarity to its final state with white lines. For both models, we observe bottom-up learning dynamics where layers closer to the input solidify into their final states faster than very top layers, which is in accord with previous studies [37, 43].

generated from CIFAR-100 dataset [23], each consisting of 10 classes. We employ the ResNet18 model [12], training only the filter atoms while keeping the atom coefficients fixed on each task. The CCA and filter subspace similarity are calculated among 45 pairs of models. The correlation between CCA and filter subspace similarity is *0.9327*, as depicted in Figure 2(b). Furthermore, the correlation between Centered Kernel Alignment (CKA) and filter subspace similarity is also reported in Table 2. These findings clearly indicate that the proposed filter subspace similarity exhibits a strong linear relationship with well-established probing-based similarities, supporting the claims made in Theorem 2.5 and Theorem 2.7.

**Limitations of probing-based similarities.** The consistency of probing-based similarities can vary depending on the probing data. Ideally, we anticipate a high similarity value when comparing models trained on the same dataset. To investigate this, we conduct an experiment where models are trained on the CIFAR-100 dataset. Figure 3(b) displays the distribution of model similarity with different probing data, where the y-axis represents the similarity and the x-axis represents the corresponding density of models. And with CIFAR-100 probing data, the CCA similarity between models yields a value over 0.8. However, when the probing data are derived from the other datasets including CIFAR-10, SVHN [38], CelebA [31], and etc., the CCA similarity drops to 0.59. A similar inconsistency in values is observed with the CKA similarity using different probing data. In contrast, the average of our proposed filter subspace similarity between models is 0.91, which aligns well with our expectation of high similarity. This finding demonstrates the effectiveness of our approach in capturing the inherent similarities between models trained on the same dataset, irrespective of the specific choice of probing data.

## 3.2 Learning Dynamics

The filter subspace similarity has various applications in analyzing NNs. It is capable of reflecting the data similarity and measuring the evolution of model similarity during the training time. We examine the training dynamics based on the heat map of filter subspace similarities. In this experiment, AlexNet [24] is trained on CIFAR-100 [23] for 150 epochs and VGG11 [50] is fine-tuned on ImageNet [47] for 20 epochs. For both models, we train and store atoms at each epoch. Figure 4 shows heat maps of similarities of the model among different training epochs.

Figure 4(a-c) are heat maps of the 1st, 3rd and 5th convolutional layers of Alexnet. We mark the epoch when the parameters of each layer reaches 0.99 similarity with the their states in the last

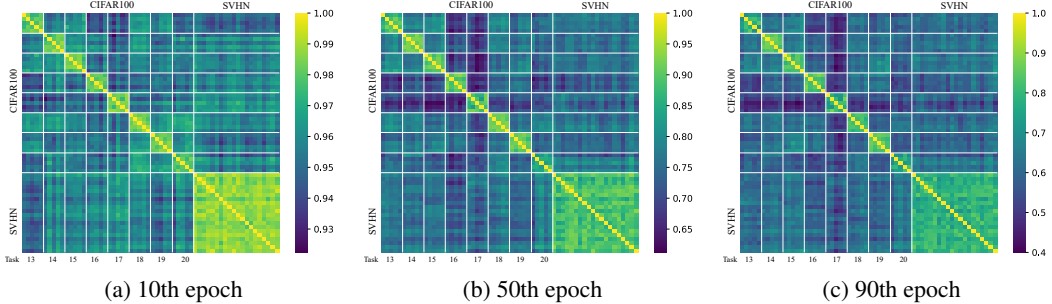

| (a) 10th epoch | (b) 50th epoch | (c) 90th epoch |

Figure 5: Similarity matrices that show relations among 60 users in FL with our filter subspace similarity through the training process. The labels of x-axis represent the ID's of CIFAR tasks. We can clearly see user clusters in all three figures. Specifically, the last 20 clients with SVHN data show higher similarities with themselves than the first 40 clients with CIFAR data, while every five of the first 40 clients sharing the same CIFAR task also show high similarities within themselves.

epoch. The first layer reaches 0.99 similarity at epoch 36 which is earlier than final layers. In Figure 4(d-f), VGG11 shows a similar behavior. Several previous works have also indicated this bottom-up learning dynamics where layers closer to the input solidify into their final states faster than very top layers [37, 43]. Our filter subspace similarity provides a highly efficient way to examine the training dynamics while showing results in accord with previous studies. Moreover, we can apply our method to calculate the similarity of a model trained on different tasks, so we can track the process of the same model interacting with different datasets. The details are shown in Appendix A.2.

## 3.3 Federated Learning

Federated learning (FL) aims at learning models collaboratively by leveraging the local computational power and data of all users with the concern of privacy [34]. Personalized Federated Learning (PFL) emerges to address some challenges in FL, such as poor convergence on heterogeneous data and lack of solution personalization [53].

In this setting, our framework achieves personalization by enforcing FL models with the shared atom coefficients for all users and specific filter atoms for each user. As illustrated in Figure 6, the shared coefficients preserve common knowledge, while user-specific atoms hold personalized information about each user. Then, we can assess model relationships with our filter subspace similarity without any probing data, which meets the privacy requirement of the FL scenario.

The shared atom coefficients can be achieved in different ways. With our framework, the coefficient can be obtained from a model pre-trained on a public dataset or from a global model trained by other FL approaches. We can also get the coefficients by training the model locally and evolving the coefficients at each communication round.

**Measuring user similarity.** With the shared atom coefficients and user-specific filter atoms, we can simply get relations of users by calculating filter subspace similarity. To be specific, we expect that users with similar data have a higher similarity. In this experiment, we distribute data of CIFAR-100 [23] and SVHN [38] to 120 clients, containing 20 SVHN clients and 100 CIFAR clients. Specifically, the SVHN dataset is randomly distributed in 20 SVHN clients. And the CIFAR-100 dataset is split into 20 subtasks with 5 classes in each subtask, and each subtask is shared by 5 CIFAR clients. The model is AlexNet [24] with 3 convolutional layers. The models share the same random initialization and filter atoms are trained independently without communication with other clients. All models are trained for $T = 100$ communication rounds on datasets. At each round, the client executes 1 epoch of SGD with momentum to train the local model, the learning rate is 0.01 and the momentum is 0.9. The experimental details are described in Appendix A.2.

Figure 5 shows the filter subspace similarity among the last 40 clients of the CIFAR-100 task and 20 clients of the SVHN task. Specifically, a distinct cluster of the 20 SVHN clients is observed, indicating a higher similarity among these clients and dissimilarity with the CIFAR clients. Additionally, every group of 5 CIFAR clients, who share the same task, also exhibit a high similarity among themselves.

Table 1: Classification accuracy of model ensemble using different FL methods and model selection strategies: Models are selected with different similarity measures in each setting. The model ensemble using our filter subspace-based method is millions of times faster and consumes much fewer resources than probing-based methods while producing comparable performance.

| FL Results | Base | +Ours | +CCA [43] | +CKA [21] |
|---|---|---|---|---|
| FedAvg [34] | $83.78 \pm 0.08$ | $\mathbf{85.82 \pm 0.35}$ | $85.65 \pm 0.21$ | $85.29 \pm 0.18$ |
| Ditto [27] | $82.98 \pm 0.13$ | $85.49 \pm 0.21$ | $\mathbf{85.54 \pm 0.19}$ | $85.37 \pm 0.2$ |
| FedRep [8] | $76.44 \pm 0.06$ | $\mathbf{78.35 \pm 0.24}$ | $78.18 \pm 0.18$ | $77.73 \pm 0.19$ |
| FedProx [29] | $80.6 \pm 0.1$ | $\mathbf{82.95 \pm 0.16}$ | $82.55 \pm 0.19$ | $82.86 \pm 0.16$ |
| FedPer [2] | $83.57 \pm 0.07$ | $\mathbf{85.21 \pm 0.2}$ | $84.91 \pm 0.18$ | $84.9 \pm 0.14$ |
| Pretrain | $81.77 \pm 0.08$ | $85.41 \pm 0.19$ | $85.24 \pm 0.13$ | $\mathbf{86.33 \pm 0.14}$ |
| *Similarity Computation Cost* | | | | |
| GFLOPs | | $\mathbf{0.019}$ | 258,610 | 2,225 |
| Time (s) | | $\mathbf{0.016}$ | 1930.4 | 92.6 |
| GPU Memory (MB) | | $\mathbf{0}$ | 4915 | 3965 |

Table 2: Continual Learning Results. The model ensemble using our filter subspace similarity is significantly faster and consumes much fewer resources than probing-based methods, while maintaining comparable classification accuracy.

| Method | CIFAR-100 | Similarity Computation Cost | | |
|---|---|---|---|---|
| | | MFLOPs | Time (s) | GPU Memory (MB) |
| AtomCL (base) | $78.11 \pm 0.13$ | - | - | - |
| +CCA [43] | $79.83 \pm 0.04$ | 35.2 | 0.26 | 1996 |
| +CKA [21] | $80.01 \pm 0.06$ | 111 | 0.3 | 1637 |
| **+Ours** | $\mathbf{80.19 \pm 0.09}$ | $\mathbf{0.007}$ | $\mathbf{0.0008}$ | $\mathbf{0}$ |

This clustering capability holds great potential for facilitating efficient cluster identification in federated learning scenarios [53]. Refer to Appendix Figure 7 for the results of all 120 clients.

The computational cost of three different approaches is shown in Figure 3(a). Notably, calculating the filter subspace similarity is significantly faster (*million* times), requiring *0* GPU memory usage than probing-based methods. Note that the advantages in computational efficiency of filter subspace similarity become more prominent as the number of models increases.

**Improving personalized model with ensemble of similar users.**  Once we get the relationships of users, we can further improve the accuracy of the current model by the ensemble of similar models, which is effective to mitigate the data heterogeneity problem in FL. The experiment is described in detail in Appendix A.2. The final results are shown in Table 1. With ensemble, the accuracies of all FL methods can be improved. Note that the results of model ensemble selected by our filter subspace similarity are comparable with probing-based methods while consuming much fewer resources.

## 3.4   Continual Learning

Continual learning is an open problem in machine learning in which data from multiple tasks arrive sequentially and the model is learned to adapt to new tasks while not forgetting the knowledge from the past [40]. Some of the tasks in continual learning are related, so models trained with these tasks can be benefited from aggregating knowledge from each other. We adopt the setting in [35], and apply filter subspace similarity to find related models. Specifically, we *10-Split* CIFAR-100 dataset, where the 100 classes is broken down into 10 tasks with 10 classes per task. We train AlexNet including atoms and atom coefficients on the first task, and train only the atoms on the following tasks. Then, we calculate the task similarity with filter subspace similarity, and report the model ensemble result with most similar members. The accuracy and the similarity computation costs are shown in Table 2. Our method provides higher results and has faster speed compared with probing-based methods.

# 4   Related Work

**Model similarity.**   Representational similarity analysis (RSA) [22] demonstrates the method of understanding brain activities by computing similarities between brain responses in different regions. Measuring the similarity of models is beneficial for understanding neural network (NN) architectures and learning dynamics  [9, 21, 37, 43]. Model similarity can be used to understand or incorporate various machine learning paradigms across different areas, including contrastive learning [13, 15], knowledge distillation [52], meta-learning [42], and transfer learning [5, 39, 44].

Multiple approaches are proposed to estimate the representational similarity of NNs. Some early works show that individual neurons can capture meaningful information [3, 4, 61, 65]. Later, gradient-based methods emerge to provide a visual explanation of deep neural networks [49]. Current popular representational similarity methods rely on features of NN. [43] proposes SVCCA to measure similarity by calculating the covariance matrix of the features of each layer after channel alignments. [21] discusses the invariance properties of similarity indices and proposes CKA with consistent correspondences between layers. probing-based similarities are data-dependent and computationally expensive. But our method measures the representational similarity only via atoms, a portion of model parameters, which is data-agnostic and much more efficient.

**Learning paradigm with numerous models.**   Some machine learning tasks involve numerous models. For example, in Federated learning [53], thousands of models are trained across clients. In Continual learning, there are multiple models generated across time [18]. Federated learning (FL) aims to improve the performance of the system by continuously training and aggregating models from users without collecting data [20, 34, 51]. FL requires communication efficiency while thousands or even millions of clients may be involved [28]. It also required to achieve personalization [14, 53] considering data heterogeneity of different users [6, 17, 28]. Estimating user similarity can effectively address these challenges in FL. Continual learning (CL) aims at providing long-term knowledge accumulation, and the main challenge is to avoid catastrophic forgetting by learning new tasks while remembering the old ones [1, 18, 19, 26, 62]. One promising way is to store neural networks for each task [16, 30, 32, 48, 60]. As the number of tasks increases, a large number of models are generated and stored. It is important to find a way to access their relations to reuse models.

**Filter atom decomposition.**   The research in task subspace modeling treats tasks as compositions of latent basis tasks and their linear combinations [10, 25, 33, 45, 64]. In the context of convolutional filter decomposition, DCFNet [41] introduces the filter subspace as an expansion of convolutional filters using a predetermined set of filter atoms. With the filter subspace, a group of tasks are separately modeled using neural networks, with sets of filter atoms learned for individual tasks, while a common set of atom coefficients is shared among tasks. The applications of filter subspace span among various domains, including domain adaptation [54, 57], continual learning [35], adaptive convolution [56, 59], image generation [55, 58], video comprehension [36], and graph convolution [7].

# 5   Conclusion

In this paper, we proposed a new paradigm for reducing representational similarity analysis in CNNs to filter subspace distance assessment, which is targeted for application scenarios where numerous models are learned. The proposed approach is targeted for application scenarios where numerous models are learned, and a computationally efficient method to assess model similarities is critical in these scenarios. We provided both theoretical and empirical evidence that the proposed filter subspace-based similarity exhibits a strong linear correlation with popular probing-based metrics while being significantly more efficient and robust in probing data. It was evaluated on both federated learning and continual learning tasks and achieves competitive performance with millions of times reduction in computational cost.

The majority of approaches in FL or CL are applied to the models with the same architecture, the proposed similarity measure with shared atom coefficient is more advantageous to be incorporated in these tasks. Our method currently assumes respective layers among compared CNNs to have coefficients with the same dimension. For our future work, we will explore the way to share atom coefficients among layers to achieve filter subspace similarity with different dimensions.

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

# A Appendix

## A.1 Theoretical Proofs

**Notations of Convolutional Operations.** In our paper, we express convolution operation as $\mathbf{Z}_u = \boldsymbol{\alpha}\mathbf{X}_p\mathbf{D}_u$. More explicitly, the formulation writes as $\mathbf{Z}_u = \boldsymbol{\alpha}\mathbf{X}_p \star \mathbf{D}_u$, where $\star$ is the convolutional operation. By converting convolutional kernel $\mathbf{D}_u$ into a Toeplitz matrix, we can replace the convolution operation $\mathbf{X}_p \star \mathbf{D}_u$ with matrix multiplication $\mathbf{X}_p\mathbf{D}_u$. We also modify $\alpha$ by $I_{hw} \bigotimes \alpha$, where $\bigotimes$ is Kronecker product, to enable the matrix multiplication $\boldsymbol{\alpha}\mathbf{X}_p\mathbf{D}_u$.

**Proposition A.1.** *Suppose $\mathbf{D}_u$ and $\mathbf{D}_v$ are two different sets of filter atoms for a convolutional layer with the common atom coefficients $\boldsymbol{\alpha}$, we can upper bound the changes in the corresponding features $\mathbf{Z}_u, \mathbf{Z}_v$ with atom changes,*

$$||\mathbf{Z}_u - \mathbf{Z}_v||_F \leq (||\boldsymbol{\alpha}||_F \lambda)\sqrt{|\mathcal{B}|} \cdot ||(\mathbf{D}_u - \mathbf{D}_v)||_F, \quad with \; \lambda = \sup_{b \in \mathcal{B}} ||\mathbf{X}||_{F,N_b}, \tag{8}$$

*Proof.* Recall the decomposed convolution can be expressed as,

$$\mathbf{Z} = \sum_{i=1}^{m} \boldsymbol{\alpha}_i \langle \mathbf{X}, \mathbf{D}[i] \rangle_{N_b} \tag{9}$$

$\forall b$ we have,

$$
\begin{aligned}
|\mathbf{Z}_u(b) - \mathbf{Z}_v(b)| &= |\sum_{i=1}^{m} \boldsymbol{\alpha}_i \langle \mathbf{X}, \mathbf{D}_u[i] \rangle_{N_b} - \sum_{i=1}^{m} \boldsymbol{\alpha}_i \langle \mathbf{X}, \mathbf{D}_v[i] \rangle_{N_b}| \\
&\leq ||\boldsymbol{\alpha}||_F (\sum_{i=1}^{m} |\langle \mathbf{X}, (\mathbf{D}_u[i] - \mathbf{D}_v[i]) \rangle_{N_b}|^2)^{1/2}.
\end{aligned}
\tag{10}
$$

By Cauchy-Schwarz inequality,

$$
\begin{aligned}
|\langle \mathbf{X}, (\mathbf{D}_u[i] - \mathbf{D}_v[i]) \rangle_{N_b}| &\leq ||\mathbf{X}||_{F,N_b} \cdot ||\mathbf{D}_u[i] - \mathbf{D}_v[i]||_{F,N_b} \\
&\leq \lambda \cdot ||\mathbf{D}_u[i] - \mathbf{D}_v[i]||_{F,N_b}
\end{aligned}
\tag{11}
$$

we have that

$$
\begin{aligned}
\sum_{b \in \mathcal{B}} |\mathbf{Z}_u(b) - \mathbf{Z}_v(b)|^2 &\leq ||\boldsymbol{\alpha}||_F^2 \sum_b \sum_{i=1}^{m} |\langle \mathbf{X}, (\mathbf{D}_u[i] - \mathbf{D}_v[i]) \rangle_{N_b}|^2 \\
&\leq ||\boldsymbol{\alpha}||_F^2 \sum_b \sum_{i=1}^{m} ||\mathbf{X}||_{F,N_b}^2 \cdot ||(\mathbf{D}_u[i] - \mathbf{D}_v[i])||_{F,N_b}^2 \\
&\leq (||\boldsymbol{\alpha}||_F \lambda)^2 \sum_{b,i} ||(\mathbf{D}_u[i] - \mathbf{D}_v[i])||_{F,N_b}^2
\end{aligned}
\tag{12}
$$

and observe that

$$\sum_{b,i} ||(\mathbf{D}_u[i] - \mathbf{D}_v[i])||_{F,N_b}^2 = \sum_{b \in \mathcal{B}} \sum_{i=1}^{m} ||(\mathbf{D}_u[i] - \mathbf{D}_v[i])||_{F,N_b}^2 = |\mathcal{B}| \cdot ||(\mathbf{D}_u - \mathbf{D}_v)||_F^2, \tag{13}$$

where $|\mathcal{B}|$ is the area of the domain of $\mathbf{X}$. Then Eq. 12 becomes

$$\sum_{b \in \mathcal{B}} |\mathbf{Z}_u(b) - \mathbf{Z}_v(b)|^2 \leq (||\boldsymbol{\alpha}||_F \lambda)^2 |\mathcal{B}| \cdot ||(\mathbf{D}_u - \mathbf{D}_v)||_F^2, \tag{14}$$

which proves that $||\mathbf{Z}_u - \mathbf{Z}_v||_F \leq (||\boldsymbol{\alpha}||_F \lambda)\sqrt{|\mathcal{B}|} \cdot ||(\mathbf{D}_u - \mathbf{D}_v)||_F$ as claimed.

$\square$

**Proposition A.2.** *Assume filter atoms $\mathbf{D}_u, \mathbf{D}_v$ are orthogonal matrices, then $\mathcal{S}_{Gras} = \mathcal{S}_{Atom}$.*

*Proof.* Since $\mathbf{D}_u, \mathbf{D}_v \in \mathbb{R}^{k^2 \times m}$ are orthogonal matrices, i.e., $\mathbf{D}_u^T \mathbf{D}_u = \mathbf{D}_v^T \mathbf{D}_v = I$, the Grassmann similarity can be represented as,

$$\mathcal{S}_{Gras}(\mathcal{F}_u, \mathcal{F}_v) = \frac{1}{m} \sum_i^m \mathbf{cos}\theta_i = \frac{1}{m} \sum_i^m \sigma_i, \tag{15}$$

where $\sigma_i = \Sigma_{ii}, U\Sigma V = \mathbf{D}_u^T \mathbf{D}_v$.

$\mathcal{S}_{Atom}$ is defined as,

$$\mathcal{S}_{Atom}(\mathcal{F}_u, \mathcal{F}_v) = \mathbf{cos}(\mathbf{D}_u, \mathbf{D}_v) = \frac{< vec(\mathbf{D}_u), vec(\mathbf{D}_v) >}{||vec(\mathbf{D}_u)||_F \cdot ||vec(\mathbf{D}_v)||_F}. \tag{16}$$

Analyze each part separately, we have $< vec(\mathbf{D}_u), vec(\mathbf{D}_v) > = \mathbf{Tr}(\mathbf{D}_u^T \mathbf{D}_v) = \sum_i^m \sigma_i$, $||vec(\mathbf{D}_u)||_F = \sqrt{\mathbf{Tr}(\mathbf{D}_u^T \mathbf{D}_u)} = \sqrt{\mathbf{Tr}(I)} = \sqrt{m}$, and also $||vec(\mathbf{D}_v)||_F = \sqrt{m}$. In total, the filter subspace similarity becomes,

$$\mathcal{S}_{Atom}(\mathcal{F}_u, \mathcal{F}_v) = \mathbf{cos}(\mathbf{D}_u, \mathbf{D}_v) = \frac{\sum_i^m \sigma_i}{m}, \tag{17}$$

which equals $\mathcal{S}_{Gras}$. The claimed theorem is proved.

$\square$

**Lemma A.3.** *For two positive semidefinite matrices* $\mathbf{A}, \mathbf{B}$,

$$\mathbf{Tr}(\mathbf{A}\mathbf{B}) \geq \sigma_{min}(\mathbf{A})\mathbf{Tr}(\mathbf{B}), \tag{18}$$

*where* $\sigma_{min}$ *denotes the minimum eigenvalue of* $A$.

*Proof.* It is equivalent to prove that,

$$\mathbf{Tr}((\mathbf{A} - \sigma_{min}(\mathbf{A})\mathbf{I})\mathbf{B}) \geq 0. \tag{19}$$

Let $\mathbf{C}, \mathbf{D}$ be matrices such that $\mathbf{A} - \sigma_{min}(\mathbf{A})\mathbf{I} = \mathbf{C}^\mathsf{T}\mathbf{C}, \mathbf{B} = \mathbf{D}^\mathsf{T}\mathbf{D}$, then

$$\begin{aligned} \mathbf{Tr}((\mathbf{A} - \sigma_{min}(\mathbf{A})\mathbf{I})\mathbf{B}) &= \mathbf{Tr}(\mathbf{C}^\mathsf{T}\mathbf{C}\mathbf{D}^\mathsf{T}\mathbf{D}) \\ &= \mathbf{Tr}(\mathbf{C}\mathbf{D}^\mathsf{T}\mathbf{D}\mathbf{C}^\mathsf{T}) \\ &= \mathbf{Tr}((\mathbf{D}\mathbf{C}^\mathsf{T})^\mathsf{T}(\mathbf{D}\mathbf{C}^\mathsf{T})) \geq 0. \end{aligned} \tag{20}$$

$\square$

**Theorem A.4.** *Suppose the forward of decomposed convolution layer for the* $u$-*th model is* $\mathbf{Z}_u = \alpha\mathbf{X}\mathbf{D}_u$. $\mathbf{Z}_u, \mathbf{Z}_v$ *nearly have zero-mean since* $\mathbf{X}_p$ *is preprocessed to be normalized. CCA coefficient is defined as* $S(\mathbf{Z}_u, \mathbf{Z}_v) = \sqrt{\frac{1}{c} \sum_{i=1}^c \sigma_i^2}$, *where* $\sigma_i^2$ *denotes the* $i$-*th eigenvalue of* $\Lambda_{u,v} = Q_u{}^\mathsf{T} Q_v$, $Q_u = \mathbf{Z}_u(\mathbf{Z}_u^\mathsf{T}\mathbf{Z}_u)^{-\frac{1}{2}}$. *Then* $\mathcal{S}(\mathbf{Z}_u, \mathbf{Z}_v)$ *is upper bounded,*

$$\mathcal{S}(\mathbf{Z}_u, \mathbf{Z}_v) \leq \frac{c^{\frac{3}{2}}\mathcal{T}}{\mathcal{C}} \mathbf{cos}(\mathbf{D}_u, \mathbf{D}_v), \tag{21}$$

where $\mathcal{T} = \mathbf{Tr}(\mathbf{X}^\mathsf{T}\boldsymbol{\alpha}^\mathsf{T}\boldsymbol{\alpha}\mathbf{X}), \mathcal{C} = \sigma_{min}(\mathbf{X}^\mathsf{T}\boldsymbol{\alpha}^\mathsf{T}\boldsymbol{\alpha}\mathbf{X})$.

*Proof.* Consider $\mathcal{S}^2 = \frac{1}{c} \sum_{i=1}^c \sigma_i^2$.

$$\mathcal{S}^2 = \frac{1}{c} \sum_{i=1}^c \sigma_i^2 = \frac{1}{c}\mathbf{Tr}(\Lambda_{u,v}\Lambda_{u,v}^\mathsf{T}). \tag{22}$$

where

$$\mathbf{Tr}(\Lambda_{u,v}\Lambda_{u,v}^\mathsf{T}) = \mathbf{Tr}(Q_u^\mathsf{T} Q_v Q_v^\mathsf{T} Q_u) = \mathbf{Tr}(Q_v Q_v^\mathsf{T} Q_u Q_u^\mathsf{T}). \tag{23}$$

As defined above, we have

$$Q_u Q_u^\intercal = \mathbf{Z}_u (\mathbf{Z}_u^\intercal \mathbf{Z}_u)^{-\frac{1}{2}} (\mathbf{Z}_u^\intercal \mathbf{Z}_u)^{-\frac{1}{2}} \mathbf{Z}_u^\intercal = \mathbf{Z}_u (\mathbf{Z}_u^\intercal \mathbf{Z}_u)^{-1} \mathbf{Z}_u^\intercal$$
$$Q_v Q_v^\intercal = \mathbf{Z}_v (\mathbf{Z}_v^\intercal \mathbf{Z}_v)^{-\frac{1}{2}} (\mathbf{Z}_v^\intercal \mathbf{Z}_v)^{-\frac{1}{2}} \mathbf{Z}_v^\intercal = \mathbf{Z}_v (\mathbf{Z}_v^\intercal \mathbf{Z}_v)^{-1} \mathbf{Z}_v^\intercal. \tag{24}$$

Then Equation 23 becomes,

$$\mathbf{Tr}(\Lambda_{u,v} \Lambda_{u,v}^\intercal) = \mathbf{Tr}(\mathbf{Z}_u (\mathbf{Z}_u^\intercal \mathbf{Z}_u)^{-1} \mathbf{Z}_u^\intercal \mathbf{Z}_v (\mathbf{Z}_v^\intercal \mathbf{Z}_v)^{-1} \mathbf{Z}_v^\intercal)$$
$$= \mathbf{Tr}((\mathbf{Z}_u^\intercal \mathbf{Z}_u)^{-1} \mathbf{Z}_u^\intercal \mathbf{Z}_v (\mathbf{Z}_v^\intercal \mathbf{Z}_v)^{-1} \mathbf{Z}_v^\intercal \mathbf{Z}_u). \tag{25}$$

By Cauchy-Schwartz Inequality,

$$\mathbf{Tr}(\Lambda_{u,v} \Lambda_{u,v}^\intercal) \leq \mathbf{Tr}((\mathbf{Z}_u^\intercal \mathbf{Z}_u)^{-1}) \mathbf{Tr}((\mathbf{Z}_v^\intercal \mathbf{Z}_v)^{-1}) \mathbf{Tr}(\mathbf{Z}_u^\intercal \mathbf{Z}_v)^2. \tag{26}$$

Then we analyze these terms individually,

$$\mathbf{Tr}(\mathbf{Z}_u^\intercal \mathbf{Z}_v) = \mathbf{Tr}(\mathbf{D}_u^\intercal \mathbf{X}^\intercal \boldsymbol{\alpha}^\intercal \boldsymbol{\alpha} \mathbf{X} \mathbf{D}_v) = \mathbf{Tr}(\mathbf{X}^\intercal \boldsymbol{\alpha}^\intercal \boldsymbol{\alpha} \mathbf{X} \mathbf{D}_v \mathbf{D}_u^\intercal)$$
$$\leq \mathbf{Tr}(\mathbf{X}^\intercal \boldsymbol{\alpha}^\intercal \boldsymbol{\alpha} \mathbf{X}) \mathbf{Tr}(\mathbf{D}_u^\intercal \mathbf{D}_v) \leq \mathcal{T} \cdot \mathbf{Tr}(\mathbf{D}_u^\intercal \mathbf{D}_v) \tag{27}$$

As for $\mathbf{Tr}((\mathbf{Z}_u^\intercal \mathbf{Z}_u)^{-1})$, let $\lambda_1, \lambda_2, ..., \lambda_c$ be eigenvalues for $\mathbf{Z}_u^\intercal \mathbf{Z}_u$ listed in descending order ($\lambda_1 \geq \lambda_2 \geq ... \geq \lambda_c$), and assume the condition number of $\mathbf{Z}_u^\intercal \mathbf{Z}_u$ and $\mathbf{Z}_v^\intercal \mathbf{Z}_v$ satisfy $\lambda_{max}/\lambda_{min} \leq \gamma$, then,

$$\mathbf{Tr}((\mathbf{Z}_u^\intercal \mathbf{Z}_u)^{-1}) = \sum_{i=1}^{c} \frac{1}{\lambda_i} \leq c \cdot \frac{1}{\lambda_c} \leq \frac{\gamma c}{\lambda_1}, \tag{28}$$

where $\lambda_1 = ||\mathbf{Z}_u^\intercal \mathbf{Z}_u||_2$, $|| \cdot ||_2$ denotes the operator norm induced by the vector $L_2$-norm. With the norm inequalities of any positive semidefinite matrix $A$,

$$||A||_2 \geq \frac{1}{\sqrt{c}} ||A||_F \geq \frac{1}{c} ||A||_* \geq \frac{1}{c} \mathbf{Tr}(A), \tag{29}$$

where $|| \cdot ||_F, || \cdot ||_*$ denote the Frobenius norm and the nuclear norm, respectively.

Equation (30) then becomes,

$$\mathbf{Tr}((\mathbf{Z}_u^\intercal \mathbf{Z}_u)^{-1}) \leq c \cdot \frac{1}{||\mathbf{Z}_u^\intercal \mathbf{Z}_u||_2} \leq \frac{\gamma c^2}{\mathbf{Tr}(\mathbf{Z}_u^\intercal \mathbf{Z}_u)}. \tag{30}$$

By Lemma A.3,
$$\mathbf{Tr}(\mathbf{Z}_u^\intercal \mathbf{Z}_u) = \mathbf{Tr}(\mathbf{D}_u^\intercal \mathbf{X}^\intercal \boldsymbol{\alpha}^\intercal \boldsymbol{\alpha} \mathbf{X} \mathbf{D}_u)$$
$$= \mathbf{Tr}(\mathbf{X}^\intercal \boldsymbol{\alpha}^\intercal \boldsymbol{\alpha}^\intercal \mathbf{X} \mathbf{D}_u \mathbf{D}_u^\intercal)$$
$$\geq \sigma_{min}(\mathbf{X}^\intercal \boldsymbol{\alpha}^\intercal \boldsymbol{\alpha}^\intercal \mathbf{X}) \mathbf{Tr}(\mathbf{D}_u^\intercal \mathbf{D}_u) \tag{31}$$
$$\geq \mathcal{C} \cdot \mathbf{Tr}(\mathbf{D}_u^\intercal \mathbf{D}_u)$$
$$\geq \mathcal{C} \cdot ||vec(\mathbf{D}_u)||_2^2,$$

where $vec(\cdot)$ denotes vectorization of a matrix.

Then Equation 30 is further derived as,

$$\mathbf{Tr}((\mathbf{Z}_u^\intercal \mathbf{Z}_u)^{-1}) \leq \frac{\gamma c^2}{\mathcal{C} \cdot ||vec(\mathbf{D}_u)||_2^2}. \tag{32}$$

Similarly, we have

$$\mathbf{Tr}((\mathbf{Z}_v^\intercal \mathbf{Z}_v)^{-1}) \leq \frac{\gamma c^2}{\mathcal{C} \cdot ||vec(\mathbf{D}_v)||_2^2}. \tag{33}$$

Finally, with $\mathbf{Tr}(\mathbf{D}_u^\mathsf{T}\mathbf{D}_v) = <vec(\mathbf{D}_u), vec(\mathbf{D}_v)>$, we have

$$
\begin{aligned}
\mathbf{Tr}(\Lambda_{u,v}\Lambda_{u,v}^\mathsf{T}) &\leq \frac{\gamma^2 \mathcal{T}^2 c^4 (<vec(\mathbf{D}_u), vec(\mathbf{D}_v)>)^2}{\mathcal{C}^2 \|vec(\mathbf{D}_u)\|_2^2 \cdot \|vec(\mathbf{D}_v)\|_2^2} \\
&\leq \frac{\gamma^2 \mathcal{T}^2 c^4}{\mathcal{C}^2} \cdot \mathbf{cos}^2(\mathbf{D}_u, \mathbf{D}_v),
\end{aligned}
\tag{34}
$$

and thus,

$$
\begin{aligned}
\mathcal{S}(\mathbf{Z}_u, \mathbf{Z}_v) &= \sqrt{\frac{1}{c}\mathbf{Tr}(\Lambda_{u,v}\Lambda_{u,v}^\mathsf{T})} \\
&\leq \frac{\gamma \mathcal{T} c^{\frac{3}{2}}}{\mathcal{C}} \cdot \mathbf{cos}(\mathbf{D}_u, \mathbf{D}_v).
\end{aligned}
\tag{35}
$$

Then the claimed theorem is proved.

$\square$

**Lemma A.5.** *For two matrices $\mathbf{A}$, $\mathbf{B}$, their frobenius norm satisfies,*

$$
\|\mathbf{AB}\|_F = \|\mathbf{A}\|_F \|\mathbf{B}\|_F \sqrt{1 - \frac{\Delta_1}{\|\mathbf{A}\|_F^2 \|\mathbf{B}\|_F^2}},
\tag{36}
$$

*where $\Delta_1 = \sum_{ij}(\sum_k A_{ik}^2)(\sum_k B_{kj}^2) \cdot \sin^2(\langle A_{i:}, B_{:j}\rangle)$.*

*Proof.* According to the definition of frobenius norm $\|\mathbf{A}\|_F = \sqrt{\sum_{ij}|A_{ij}|^2}$ we have,

$$
\|\mathbf{AB}\|_F = \sqrt{\sum_{ij}(\sum_k A_{ik}B_{kj})^2}.
\tag{37}
$$

Note that $(\sum_i x_i y_i)^2 = (\sum_i x_i^2)(\sum_i y_i^2) \cdot \cos^2(\langle x, y\rangle) = (\sum_i x_i^2)(\sum_i y_i^2) - (\sum_i x_i^2)(\sum_i y_i^2) \cdot \sin^2(\langle x, y\rangle)$, where $\langle x, y\rangle$ is the angle of two vectors $x$ and $y$. We have,

$$
\begin{aligned}
&\sqrt{\sum_{ij}(\sum_k A_{ik}B_{kj})^2} \\
&= \sqrt{\sum_{ij}\left[(\sum_k A_{ik}^2)(\sum_k B_{kj}^2) - (\sum_k A_{ik}^2)(\sum_k B_{kj}^2) \cdot \sin^2(\langle A_{i:}, B_{:j}\rangle)\right]} \\
&= \sqrt{\sum_{ik}A_{ik}^2}\sqrt{\sum_{kj}B_{kj}^2}\sqrt{1 - \frac{\sum_{ij}(\sum_k A_{ik}^2)(\sum_k B_{kj}^2) \cdot \sin^2(\langle A_{i:}, B_{:j}\rangle)}{\sum_{ik}A_{ik}^2 \sum_{kj}B_{kj}^2}} \\
&= \|\mathbf{A}\|_F \|\mathbf{B}\|_F \sqrt{1 - \frac{\sum_{ij}(\sum_k A_{ik}^2)(\sum_k B_{kj}^2) \cdot \sin^2(\langle A_{i:}, B_{:j}\rangle)}{\|\mathbf{A}\|_F^2 \|\mathbf{B}\|_F^2}} \\
&= \|\mathbf{A}\|_F \|\mathbf{B}\|_F \sqrt{1 - \frac{\Delta_1}{\|\mathbf{A}\|_F^2 \|\mathbf{B}\|_F^2}},
\end{aligned}
\tag{38}
$$

where $A_{i:}$ is the $i$-th row of $\mathbf{A}$ and $B_{:j}$ is the $j$-th column of $\mathbf{B}$, $\Delta_1 = \sum_{ij}(\sum_k A_{ik}^2)(\sum_k B_{kj}^2) \cdot \sin^2(\langle A_{i:}, B_{:j}\rangle)$. As $A_{i:}$ and $B_{:j}$ are more correlated, $\langle A_{i:}, B_{:j}\rangle \to 0$, thus, $\Delta_1 \ll \|\mathbf{A}\|_F^2 \|\mathbf{B}\|_F^2$.

$\square$

**Lemma A.6.**

$$
\|\mathbf{A}^{1/2}\|_F = \|\mathbf{A}\|_F^{1/2}(1 + \frac{\Delta_{1\mathbf{A}^{1/2}}}{\|\mathbf{A}\|_F^2})^{1/4}.
\tag{39}
$$

*Proof.* According to Lemma A.5, we have,

$$\|\mathbf{A}\|_F^2 = \|\mathbf{A}^{1/2}\|_F^4 - \Delta_1. \tag{40}$$

Thus,

$$\|\mathbf{A}^{1/2}\|_F = \|\mathbf{A}\|_F^{1/2}(1 + \frac{\Delta_{1A^{1/2}}}{\|\mathbf{A}\|_F^2})^{1/4}, \tag{41}$$

where $\Delta_{1\mathbf{A}^{1/2}} = \sum_{ij}(\sum_k (A^{1/2})_{ik}^2)(\sum_k (A^{1/2})_{kj}^2) \cdot \sin^2(\langle (A^{1/2})_{i:}, (A^{1/2})_{:j}\rangle)$. As $(A^{1/2})_{i:}$ and $(A^{1/2})_{:j}$ are more correlated, $\langle (A^{1/2})_{i:}, (A^{1/2})_{:j}\rangle \to 0$, thus, $\Delta_{1A^{1/2}} \ll \|\mathbf{A}\|_F^2$.

$\square$

**Lemma A.7.** *For three matrices* $\mathbf{A}$, $\mathbf{B}$*, and* $\mathbf{C}$*, their frobenius norm satisfies,*

$$\|\mathbf{A}\|_F = \|\mathbf{A}\|_F\|\mathbf{B}\|_F\|\mathbf{C}\|_F\sqrt{1 - \frac{\Delta_2 + \Delta_3}{\|\mathbf{A}\|_F^2\|\mathbf{B}\|_F^2\|\mathbf{C}\|_F^2}}, \tag{42}$$

*where* $\Delta_2 = \frac{1}{2}[\|\mathbf{A}\|_F^2 \sum_{kj}(\sum_l B_{kl}^2)(\sum_l C_{lj}^2) \cdot \sin^2(\langle B_{k:}, C_{:j}\rangle) + \|\mathbf{C}\|_F^2 \sum_{il}(\sum_k A_{ik}^2)(\sum_k B_{kl}^2) \cdot \sin^2(\langle A_{i:}, B_{:l}\rangle)]$ *and* $\Delta_3 = \frac{1}{2}[\sum_{ij}(\sum_k A_{ik}^2)(\sum_k (BC)_{kj}^2) \cdot \sin^2(\langle A_{i:}, (BC)_{:j}\rangle) + \sum_{ij}(\sum_l (AB)_{il}^2)(\sum_l C_{lj}^2) \cdot \sin^2(\langle (AB)_{i:}, C_{:j}\rangle)]$.

*Proof.* Based on Lemma A.5, we have,

$$\begin{aligned}
&\|\mathbf{ABC}\|_F^2 \\
=&\|\mathbf{AB}\|_F^2\|\mathbf{C}\|_F^2 - \sum_{ij}(\sum_l (AB)_{il}^2)(\sum_l C_{lj}^2) \cdot \sin^2(\langle (AB)_{i:}, C_{:j}\rangle) \\
=&\|\mathbf{A}\|_F^2\|\mathbf{B}\|_F^2\|\mathbf{C}\|_F^2 - \|\mathbf{C}\|_F^2 \sum_{il}(\sum_k A_{ik}^2)(\sum_k B_{kl}^2) \cdot \sin^2(\langle A_{i:}, B_{:l}\rangle) \\
&- \sum_{ij}(\sum_l (AB)_{il}^2)(\sum_l C_{lj}^2) \cdot \sin^2(\langle (AB)_{i:}, C_{:j}\rangle)
\end{aligned} \tag{43}$$

Symmetrically, we also have,

$$\begin{aligned}
&\|\mathbf{ABC}\|_F^2 \\
=&\|\mathbf{A}\|_F^2\|\mathbf{BC}\|_F^2 - \sum_{ij}(\sum_k A_{ik}^2)(\sum_k (BC)_{kj}^2) \cdot \sin^2(\langle A_{i:}, (BC)_{:j}\rangle) \\
=&\|\mathbf{A}\|_F^2\|\mathbf{B}\|_F^2\|\mathbf{C}\|_F^2 - \|\mathbf{A}\|_F^2 \sum_{kj}(\sum_l B_{kl}^2)(\sum_l C_{lj}^2) \cdot \sin^2(\langle B_{k:}, C_{:j}\rangle) \\
&- \sum_{ij}(\sum_k A_{ik}^2)(\sum_k (BC)_{kj}^2) \cdot \sin^2(\langle A_{i:}, (BC)_{:j}\rangle)
\end{aligned} \tag{44}$$

Thus,

$$\begin{aligned}
&\|\mathbf{ABC}\|_F^2 \\
=&\frac{1}{2}[\|\mathbf{A}\|_F^2\|\mathbf{B}\|_F^2\|\mathbf{C}\|_F^2 - \|\mathbf{A}\|_F^2 \sum_{kj}(\sum_l B_{kl}^2)(\sum_l C_{lj}^2) \cdot \sin^2(\langle B_{k:}, C_{:j}\rangle) \\
&- \sum_{ij}(\sum_k A_{ik}^2)(\sum_k (BC)_{kj}^2) \cdot \sin^2(\langle A_{i:}, (BC)_{:j}\rangle) \\
&+ \|\mathbf{A}\|_F^2\|\mathbf{B}\|_F^2\|\mathbf{C}\|_F^2 - \|\mathbf{C}\|_F^2 \sum_{il}(\sum_k A_{ik}^2)(\sum_k B_{kl}^2) \cdot \sin^2(\langle A_{i:}, B_{:l}\rangle) \\
&- \sum_{ij}(\sum_l (AB)_{il}^2)(\sum_l C_{lj}^2) \cdot \sin^2(\langle (AB)_{i:}, C_{:j}\rangle)] \\
=&\|\mathbf{A}\|_F^2\|\mathbf{B}\|_F^2\|\mathbf{C}\|_F^2 - \Delta_2 - \Delta_3,
\end{aligned} \tag{45}$$

where $\Delta_2 = \frac{1}{2}[\|A\|_F^2 \sum_{kj}(\sum_l B_{kl}^2)(\sum_l C_{lj}^2) \cdot \sin^2(\langle B_{k:}, C_{:j}\rangle) + \|C\|_F^2 \sum_{il}(\sum_k A_{ik}^2)(\sum_k B_{kl}^2) \cdot \sin^2(\langle A_{i:}, B_{:l}\rangle)]$ and $\Delta_3 = \frac{1}{2}[\sum_{ij}(\sum_k A_{ik}^2)(\sum_k (BC)_{kj}^2) \cdot \sin^2(\langle A_{i:}, (BC)_{:j}\rangle) + \sum_{ij}(\sum_l (AB)_{il}^2)(\sum_l C_{lj}^2) \cdot \sin^2(\langle (AB)_{i:}, C_{:j}\rangle)]$. Therefore,

$$\|\mathbf{ABC}\|_F = \|\mathbf{A}\|_F \|\mathbf{B}\|_F \|\mathbf{C}\|_F \sqrt{1 - \frac{\Delta_2 + \Delta_3}{\|\mathbf{A}\|_F^2 \|\mathbf{B}\|_F^2 \|\mathbf{C}\|_F^2}}. \tag{46}$$

As $A_{i:}$ and $B_{:l}$, $B_{k:}$ and $C_{:j}$ are more correlated, $\langle A_{i:}, B_{:l}\rangle, \langle B_{k:}, C_{:j}\rangle, \langle A_{i:}, (BC)_{:j}\rangle, \langle (AB)_{i:}, C_{:j}\rangle \to 0$, thus, $\Delta_2 \ll \|\mathbf{A}\|_F^2 \|\mathbf{B}\|_F^2 \|\mathbf{C}\|_F^2$ and $\Delta_3 \ll \|\mathbf{A}\|_F^2 \|\mathbf{B}\|_F^2 \|\mathbf{C}\|_F^2$.

$\square$

**Lemma A.8.**

$$\|\mathbf{A}^{-1/2}\mathbf{B}\mathbf{C}^{-1/2}\|_F = \kappa_F(\mathbf{A}^{1/2})\kappa_F(\mathbf{C}^{1/2})\frac{\|\mathbf{B}\|_F}{\|\mathbf{A}^{1/2}\|_F\|\mathbf{C}^{1/2}\|_F}\sqrt{1 - \frac{\Delta_2 + \Delta_3}{\|\mathbf{A}^{-1/2}\|_F^2\|\mathbf{B}\|_F^2\|\mathbf{C}^{-1/2}\|_F^2}}, \tag{47}$$

where $\kappa_F(\mathbf{A}^{1/2})$ and $\kappa_F(\mathbf{C}^{1/2})$ are the condition number of $\mathbf{A}^{1/2}$ and $\mathbf{C}^{1/2}$, $\kappa_F(\mathbf{A}^{1/2}) = \sqrt{(\sum \sigma_i^2(\mathbf{A}^{1/2}))(\sum \frac{1}{\sigma_i^2(\mathbf{A}^{1/2})})}$ and $\kappa_F(\mathbf{C}^{1/2}) = \sqrt{(\sum \sigma_i^2(\mathbf{C}^{1/2}))(\sum \frac{1}{\sigma_i^2(\mathbf{C}^{1/2})})}$; $\sigma_i^2(\mathbf{A}^{1/2})$ are singular value of $\mathbf{A}^{1/2}$ and $\sigma_i^2(\mathbf{C}^{1/2})$ are singular value of $\mathbf{C}^{1/2}$.

*Proof.* Based on Lemma A.7, we have,

$$\|\mathbf{A}^{-1/2}\mathbf{B}\mathbf{C}^{-1/2}\|_F = \|\mathbf{A}^{-1/2}\|_F\|\mathbf{B}\|_F\|\mathbf{C}^{-1/2}\|_F\sqrt{1 - \frac{\Delta_2 + \Delta_3}{\|\mathbf{A}^{-1/2}\|_F^2\|\mathbf{B}\|_F^2\|\mathbf{C}^{-1/2}\|_F^2}}. \tag{48}$$

By the definition of condition number $\kappa_F(\mathbf{X}) = \|\mathbf{X}\|_F\|\mathbf{X}^{-1}\|_F = \sqrt{(\sum \sigma_i^2(\mathbf{X}))(\sum \frac{1}{\sigma_i^2(\mathbf{X})})}$,

$$\|\mathbf{A}^{-1/2}\mathbf{B}\mathbf{C}^{-1/2}\|_F = \kappa_F(\mathbf{A}^{1/2})\kappa_F(\mathbf{C}^{1/2})\frac{\|\mathbf{B}\|_F}{\|\mathbf{A}^{1/2}\|_F\|\mathbf{C}^{1/2}\|_F}\sqrt{1 - \frac{\Delta_2 + \Delta_3}{\|\mathbf{A}^{-1/2}\|_F^2\|\mathbf{B}\|_F^2\|\mathbf{C}^{-1/2}\|_F^2}}. \tag{49}$$

$\square$

**Theorem A.9.** *Suppose the forward of decomposed convolution layer for the $u$-th model is $\mathbf{Z}_u = \alpha \mathbf{X} \mathbf{D}_u$, CCA coefficient be $S(\mathbf{Z}_u, \mathbf{Z}_v) = \sqrt{\frac{1}{c}\sum_{i=1}^c \sigma_i^2}$, where $\sigma_i^2$ denotes the $i$-th eigenvalue of $\Lambda_{u,v} = Q_u^\mathsf{T} Q_v$, $Q_u = \mathbf{Z}_u(\mathbf{Z}_u^\mathsf{T}\mathbf{Z}_u)^{-\frac{1}{2}}$. Then $\mathcal{S}(\mathbf{Z}_u, \mathbf{Z}_v)$ is approximately linear to filter subspace similarity,*

$$\mathcal{S}(\mathbf{Z}_u, \mathbf{Z}_v) = \frac{\gamma_1\gamma_2\gamma_3}{\sqrt{c}}\cos(\mathbf{D}_u, \mathbf{D}_v), \tag{50}$$

*Proof.* Based on $S(\mathbf{Z}_u, \mathbf{Z}_v) = \sqrt{\frac{1}{c}\sum_{i=1}^c \sigma_i^2}$ and $\|\Lambda_{u,v}\|_F = \sqrt{\sum_{i=1}^c \sigma_i^2}$, where $\sigma_i$ are the singular value of $\Lambda_{u,v}$,

$$S = \sqrt{\frac{1}{c}\sum_{i=1}^c \sigma_i^2} = \frac{1}{\sqrt{c}}\|\Lambda_{u,v}\|_F = \frac{1}{\sqrt{c}}\|(\mathbf{Z}_u^\mathsf{T}\mathbf{Z}_u)^{-\frac{1}{2}}\mathbf{Z}_u^\mathsf{T}\mathbf{Z}_v(\mathbf{Z}_v^\mathsf{T}\mathbf{Z}_v)^{-\frac{1}{2}}\|_F. \tag{51}$$

According to Lemma. A.8, we have

$$\frac{1}{\sqrt{c}}\|(\mathbf{Z}_u^\mathsf{T}\mathbf{Z}_u)^{-\frac{1}{2}}\mathbf{Z}_u^\mathsf{T}\mathbf{Z}_v(\mathbf{Z}_v^\mathsf{T}\mathbf{Z}_v)^{-\frac{1}{2}}\|_F = \frac{\gamma_1\gamma_2}{\sqrt{c}}\frac{\|\mathbf{Z}_u^\mathsf{T}\mathbf{Z}_v\|_F}{\|(\mathbf{Z}_u^\mathsf{T}\mathbf{Z}_u)^{\frac{1}{2}}\|_F\|(\mathbf{Z}_v^\mathsf{T}\mathbf{Z}_v)^{\frac{1}{2}}\|_F}, \tag{52}$$

where $\gamma_1 = \kappa_F((\mathbf{Z}_u^\top \mathbf{Z}_u)^{\frac{1}{2}}) \cdot \kappa_F((\mathbf{Z}_v^\top \mathbf{Z}_v)^{\frac{1}{2}})$ and $\gamma_2 = \sqrt{1 - \frac{\Delta_2 + \Delta_3}{\|(\mathbf{Z}_u^\top \mathbf{Z}_u)^{-1/2}\|_F^2 \|\mathbf{Z}_u^\top \mathbf{Z}_v\|_F^2 \|(\mathbf{Z}_v^\top \mathbf{Z}_v)^{-1/2}\|_F^2}}$.

As $\mathbf{Z}_u = \boldsymbol{\alpha} \mathbf{X} \mathbf{D}_u$ and $\mathbf{Z}_v = \boldsymbol{\alpha} \mathbf{X} \mathbf{D}_v$, we have

$$
\begin{aligned}
&\frac{\gamma_1 \gamma_2}{\sqrt{c}} \frac{\|\mathbf{Z}_u^\top \mathbf{Z}_v\|_F}{\|(\mathbf{Z}_u^\top \mathbf{Z}_u)^{\frac{1}{2}}\|_F \|(\mathbf{Z}_v^\top \mathbf{Z}_v)^{\frac{1}{2}}\|_F} \\
&= \frac{\gamma_1 \gamma_2}{\sqrt{c}} \frac{\|\mathbf{D}_u^\top \mathbf{X}^\top \boldsymbol{\alpha}^\top \boldsymbol{\alpha} \mathbf{X} \mathbf{D}_v\|_F}{\|(\mathbf{D}_u^\top \mathbf{X}^\top \boldsymbol{\alpha}^\top \boldsymbol{\alpha} \mathbf{X} \mathbf{D}_u)^{\frac{1}{2}}\|_F \|(\mathbf{D}_v^\top \mathbf{X}^\top \boldsymbol{\alpha}^\top \boldsymbol{\alpha} \mathbf{X} \mathbf{D}_v)^{\frac{1}{2}}\|_F}.
\end{aligned}
\tag{53}
$$

According to Lemma A.6,

$$
\begin{aligned}
&\frac{\gamma_1 \gamma_2}{\sqrt{c}} \frac{\|\mathbf{D}_u^\top \mathbf{X}^\top \boldsymbol{\alpha}^\top \boldsymbol{\alpha} \mathbf{X} \mathbf{D}_v\|_F}{\|(\mathbf{D}_u^\top \mathbf{X}^\top \boldsymbol{\alpha}^\top \boldsymbol{\alpha} \mathbf{X} \mathbf{D}_u)^{\frac{1}{2}}\|_F \|(\mathbf{D}_v^\top \mathbf{X}^\top \boldsymbol{\alpha}^\top \boldsymbol{\alpha} \mathbf{X} \mathbf{D}_v)^{\frac{1}{2}}\|_F} \\
&= \frac{\gamma_1 \gamma_2 \gamma_3}{\sqrt{c}} \frac{\|\mathbf{D}_u^\top \mathbf{X}^\top \boldsymbol{\alpha}^\top \boldsymbol{\alpha} \mathbf{X} \mathbf{D}_v\|_F}{\|(\mathbf{D}_u^\top \mathbf{X}^\top \boldsymbol{\alpha}^\top \boldsymbol{\alpha} \mathbf{X} \mathbf{D}_u)\|_F^{\frac{1}{2}} \|(\mathbf{D}_v^\top \mathbf{X}^\top \boldsymbol{\alpha}^\top \boldsymbol{\alpha} \mathbf{X} \mathbf{D}_v)\|_F^{\frac{1}{2}}},
\end{aligned}
\tag{54}
$$

where $\gamma_3 = (1 + \frac{\Delta_1}{\|(\mathbf{D}_u^\top \mathbf{X}^\top \boldsymbol{\alpha}^\top \boldsymbol{\alpha} \mathbf{X} \mathbf{D}_u)\|_F^2})^{-\frac{1}{4}} (1 + \frac{\Delta_1}{\|(\mathbf{D}_v^\top \mathbf{X}^\top \boldsymbol{\alpha}^\top \boldsymbol{\alpha} \mathbf{X} \mathbf{D}_v)\|_F^2})^{-\frac{1}{4}}$.

As Assumption 2.6 holds, it becomes

$$
\begin{aligned}
&\frac{\gamma_1 \gamma_2 \gamma_3}{\sqrt{c}} \frac{\|\mathbf{D}_u^\top \mathbf{X}^\top \boldsymbol{\alpha}^\top \boldsymbol{\alpha} \mathbf{X} \mathbf{D}_v\|_F}{\|(\mathbf{D}_u^\top \mathbf{X}^\top \boldsymbol{\alpha}^\top \boldsymbol{\alpha} \mathbf{X} \mathbf{D}_u)\|_F^{\frac{1}{2}} \|(\mathbf{D}_v^\top \mathbf{X}^\top \boldsymbol{\alpha}^\top \boldsymbol{\alpha} \mathbf{X} \mathbf{D}_v)\|_F^{\frac{1}{2}}} \\
&= \frac{\gamma_1 \gamma_2 \gamma_3}{\sqrt{c}} \frac{\|\mathbf{D}_u^\top \mathbf{D}_v\|_F \|\mathbf{X}^\top \boldsymbol{\alpha}^\top \boldsymbol{\alpha} \mathbf{X}\|_F}{\|\mathbf{D}_u^\top\|_F^{\frac{1}{2}} \|\mathbf{X}^\top \boldsymbol{\alpha}^\top \boldsymbol{\alpha} \mathbf{X}\|_F^{\frac{1}{2}} \|\mathbf{D}_u\|_F^{\frac{1}{2}} \|\mathbf{D}_v^\top\|_F^{\frac{1}{2}} \|\mathbf{X}^\top \boldsymbol{\alpha}^\top \boldsymbol{\alpha} \mathbf{X}\|_F^{\frac{1}{2}} \|\mathbf{D}_v\|_F^{\frac{1}{2}}} \\
&= \frac{\gamma_1 \gamma_2 \gamma_3}{\sqrt{c}} \frac{\|\mathbf{D}_u^\top \mathbf{D}_v\|_F}{\|\mathbf{D}_u\|_F \|\mathbf{D}_v\|_F} \\
&= \frac{\gamma_1 \gamma_2 \gamma_3}{\sqrt{c}} \cos(\mathbf{D}_u, \mathbf{D}_v).
\end{aligned}
\tag{55}
$$

Thus, we have

$$
\mathcal{S}(\mathbf{Z}_u, \mathbf{Z}_v) = \frac{\gamma_1 \gamma_2 \gamma_3}{\sqrt{c}} \cos(\mathbf{D}_u, \mathbf{D}_v).
\tag{56}
$$

Specifically, we have $\gamma_2 = \sqrt{1 - \frac{\Delta}{\gamma_1^2 \gamma_3^2} \frac{1}{cos^2(\mathbf{D}_u, \mathbf{D}_v)}}$, and since $\Delta$ are small, with Taylor expansion, $\gamma_2 \approx 1 - \frac{1}{2} \frac{\Delta}{\gamma_1^2 \gamma_3^2} \frac{1}{cos^2(\mathbf{D}_u, \mathbf{D}_v)}$. The term $\frac{1}{cos^2(\mathbf{D}_u, \mathbf{D}_v)}$ causes non-linearity in the relation between CCA and filter subspace similarity. $\square$

## A.2 Experiment Settings

**Model training of Federated Learning.** In each experiment we have 100 clients in total and sample a ratio $r = 0.1$ of all the clients on every round. All models are randomly initialized and trained for $T = 100$ communication rounds for the CIFAR datasets. At each round, the client executes 15 epochs of SGD with momentum to train the local model, the learning rate is 0.01 and momentum is 0.9. Accuracies are computed by taking the average local accuracies for all users at the final communication round. As shown in the Table 3, we have different settings for CIFAR-10 and CIFAR-100. For example, $(100, 2)$ means 100 clients with 2 classes on each client. For each method, the training takes about 12 hours on Nvidia RTX A5000.

**Comparison with other FL approaches.** We compare our approach by evolving shared atom coefficients with various personalized federated learning methods and federated learning methods with local finetuning. Among these methods, FedPer [2] and FedRep[8] have the similar ideas by learning shared global representation and personalized local heads. Ditto [27] and FedProx [29] induce global regularization to improve the model performance. We also compare our method with FedAvg [34]. FedRep [8] approaches the common knowledge with shared representation. The codes

Table 3: Compare accuracy with different approaches

| (# client, # classes per client) | CIFAR-100 | | CIFAR-10 | | |
|---|---|---|---|---|---|
| | (100, 5) | (100, 20) | (100, 2) | (100, 5) | (1000, 2) |
| FedAvg | 82.39 | 62.92 | 86.37 | 70.63 | 86.12 |
| FedProx | 80.77 | 59.7 | 85.90 | 69.94 | 84.83 |
| FedPer | 81.46 | 62.52 | 81.74 | 68.24 | 81.74 |
| FedRep | 72.98 | 37.71 | 80.55 | 67.3 | 82.98 |
| Local | 81.21 | 49.25 | 90.24 | 72.05 | 97.80 |
| Ours | 81.03 | 52.13 | 83.37 | 65.63 | 82.54 |

are adapted from [†]. We evaluate the test accuracy on CIFAR-10 and CIFAR-100 with different FL setting. As shown in Table 3, our method achieves comparable performance among different methods.

**Fine-tuning models for ensemble.**    We select 3 models with different similarity measures for ensemble. For feature-based similarity methods, we randomly select 1000 examples from CIFAR-100 dataset. The fully-connected layer of each model is fine-tuned on the user's local data with 100 epochs. The fine-tuning takes about 12 hours on Nvidia RTX A5000. After fine-tuning, the accuracy is measured on local test data, with the predictions of current model and 3 selected models.

## A.3    Extra Experiments

**Representation dependency on filter atoms.**    We first validate the dependency of deep features on filter atoms in Proposition 2.1 with a simple experiment. The model $\mathcal{F}$ here is a 2-layer CNN with coefficient $\alpha$ and atom $\mathbf{D}$ generated from normal distribution $\mathcal{N}(0,1)$. The input sample $\mathbf{X}$ is also generated from normal distribution $\mathcal{N}(0,1)$. Figure 8(a) shows the relation between $\|\mathbf{Z}_u - \mathbf{Z}_v\|_F$ and $\|\mathbf{D}_u - \mathbf{D}_v\|_F$ by fixing coefficient $\alpha$ and input sample $\mathbf{X}$ and randomly varying filter atoms $\mathbf{D}$. All the points are below the line which is the bound provided by Proposition 2.1, reflecting that the representation variations are dominated by filter atoms.

**Correlation between probing-based and filter subspace-based methods.**    In addition, we empirically verify that CCA and filter subspace similarity have a strong correlation with AlexNet. In this experiment, 10 tasks are generated from CIFAR100 [23] with 10 classes in each task. Only the filter atoms of each task are trained while the atom coefficients are fixed. We calculate CCA and filter subspace similarity among 45 pairs of models. The correlation between CCA and filter subspace similarity is *0.8638* which is shown in Figure 9(b). Similarly, the correlation between CKA and filter subspace similarity is also reported in Figure 9 (Table). These results clearly show that the proposed filter subspace similarity has high linear relationship with popular probing-based similarities, which agrees with Theorem 2.5 and Theorem 2.7.

[†]`https://github.com/lgcollins/FedRep`

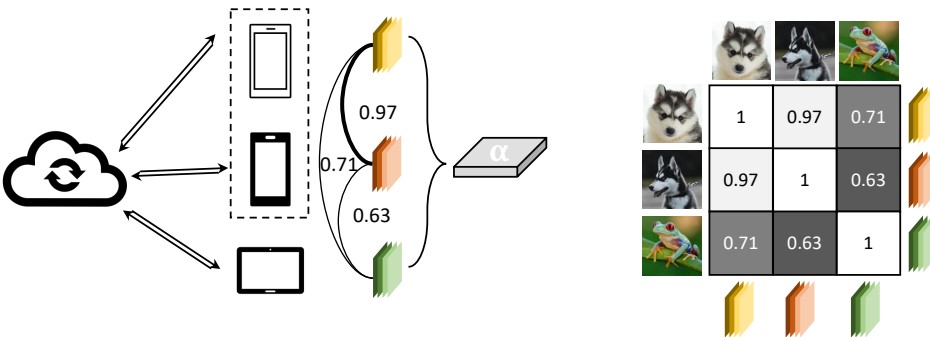

Figure 6: The shared coefficients and user-specific atoms represent common knowledge and personalized information. The filter subspace similarity is used to calculate the relations among users. Users with heterogeneous data result in lower similarity, as illustrated in a similarity matrix.

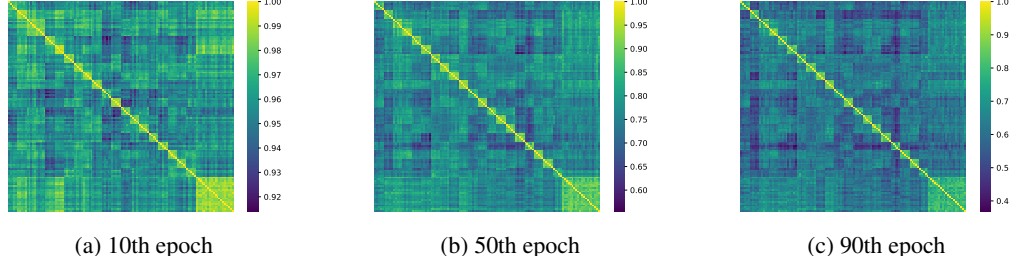

(a) 10th epoch         (b) 50th epoch         (c) 90th epoch

Figure 7: Similarity matrices that show relations among 120 users in FL with our filter subspace similarity through the training process.

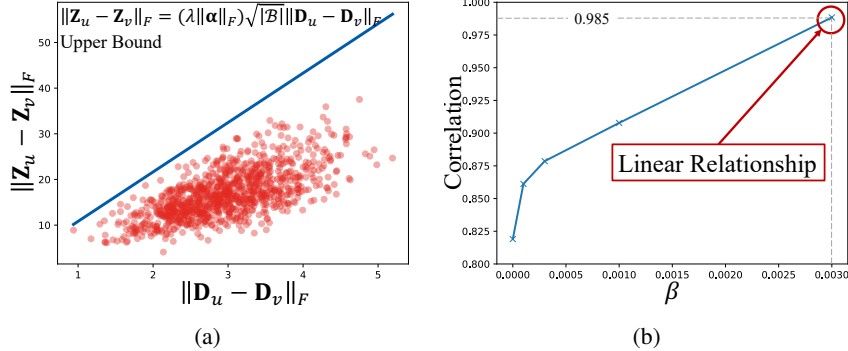

(a)                     (b)

Figure 8: (a) The change of features $\|\mathbf{Z}_u - \mathbf{Z}_v\|_F$ is bounded by the change of atoms $\|\mathbf{D}_u - \mathbf{D}_v\|_F$. (b) The channel decorrelation leads to a higher correlation between CCA and filter subspace similarity. And the correlation can reach 0.985 with $\beta = 3 \times 10^{-3}$, which means a near linear relation between CCA and filter subspace similarity.

**Effect of channel decorrelation.** We further design a regularization term $\beta \sum_{i \neq j} (\mathbf{Z}_u^\mathsf{T} \mathbf{Z}_u)_{ij}^2$ to approach $(\mathbf{Z}_u^\mathsf{T} \mathbf{Z}_u)_{ii} \gg (\mathbf{Z}_u^\mathsf{T} \mathbf{Z}_u)_{ij}$ in Assumption. 2.6. As shown in Figure 8(b), the correlation between CCA and filter subspace similarity keeps increasing as $\beta$ increases. The correlation reaches 0.985 when $\beta = 3 \times 10^{-3}$, indicating a near-linear relationship, which is aligned with Theorem. 2.7.

**Similar representations across datasets.** Similar to [21], we can use filter subspace similarity to compare networks trained on different datasets. In Figure 10(a), we show that pairs of models that are both trained on CIFAR-10 and CIFAR-100 have high atom-based similarities. Models learned on two datasets respectively still show high similarity. In contrast, similarities between trained and untrained models are significantly lower.

**Limitation of probing-based methods.** As shown in Figure 10(b), to illustrate sensitivity of probing-based similarities to probing data, we perform a simple regression task with data, $\{(x_i = 0, y_i, z_i)\}_{i=1}^n$, where $z_i = f(x_i, y_i) + \epsilon_i$ and $y_i, \epsilon_i \sim \mathcal{N}(0.5, 0.1)$. Two NN models $\mathcal{F}_1$ and $\mathcal{F}_2$

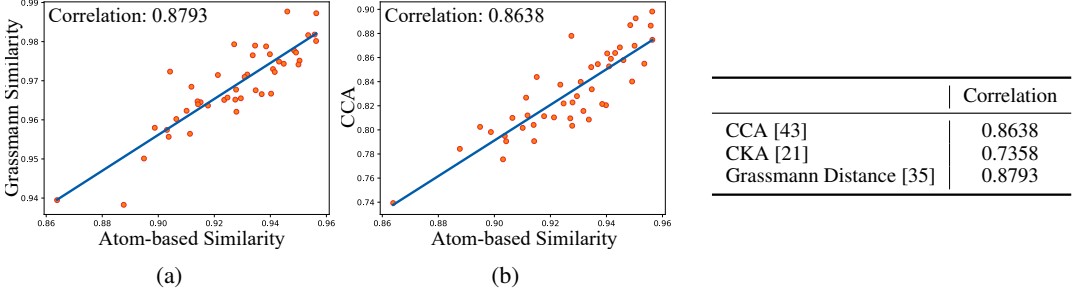

(a)                     (b)

|  | Correlation |
|---|---|
| CCA [43] | 0.8638 |
| CKA [21] | 0.7358 |
| Grassmann Distance [35] | 0.8793 |

Figure 9: (a) Correlation between Grassmann similarity and filter subspace similarity; (b) Correlation between CCA and filter subspace similarity. (Table) Correlation between filter subspace similarity and other approaches.

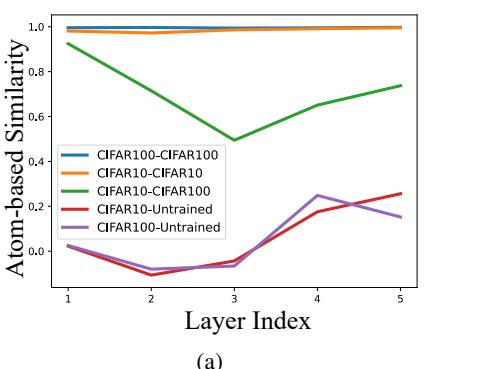 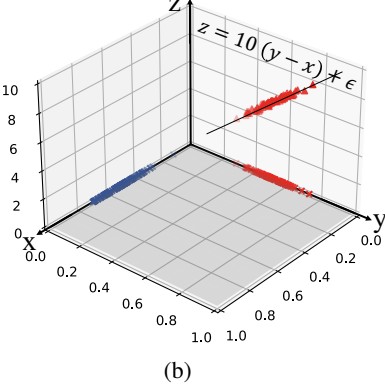

|         (a)         |         (b)         |

Figure 10: (a) Using filter subspace similarity, models trained on different datasets (CIFAR-10 and CIFAR-100) are similar among themselves, but they differ from untrained models. (b) Illustration of limitations of probing-based similarities. Input data from "red" ($\{(x_i = 0, y_i)\}$) and "blue" ($\{(x_i' = y_i, y_i' = 0)\}$) are orthogonal. Since two models are learned on "red" data, their similarity should be 1, which can be faithfully indicated by our atom similarity. However, probing-based similarities will become 0 with the "blue" probing data.

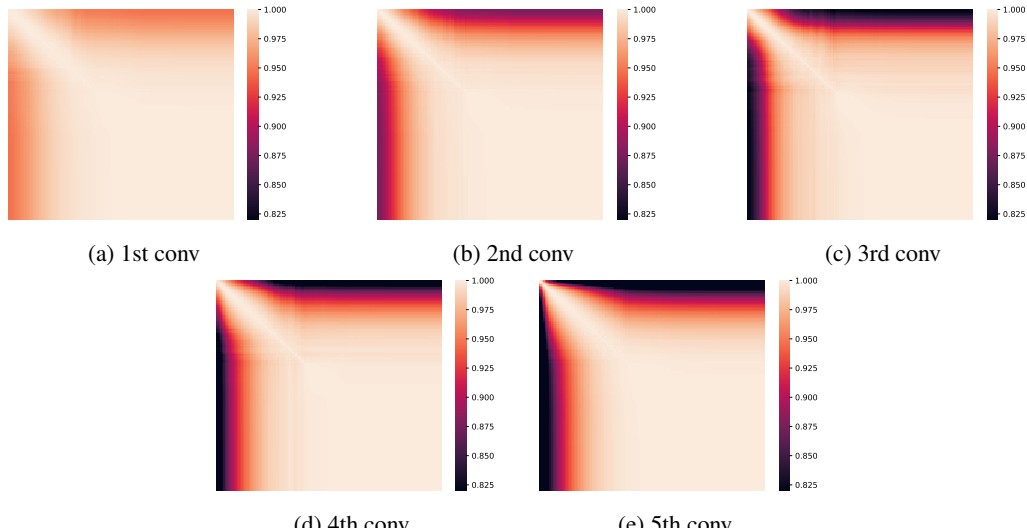

| (a) 1st conv | (b) 2nd conv | (c) 3rd conv |

| (d) 4th conv | (e) 5th conv |

Figure 11: Similarity of AlexNet with atoms from different time point during the training.

with the same initialization and atom coefficients are trained for their different atoms to learn $\mathcal{F} : (X, Y) \to Z$. It is can be simply found that the filter subspace similarity of $\mathcal{F}_1$ and $\mathcal{F}_2$ is 1 and the probing-based similarity is also 1 with the same $\{(x_i = 0, y_i)\}$ as the probing data. However, if we choose $\{(x_i' = y_i, y_i' = 0)\}$ as the probing data, then the probing-based similarities directly become **0** as the data are now orthogonal to model parameters.

### A.4 Training dynamics.

We investigate the training dynamics of AlexNet [24] and VGG [50] separately on CIFAR-100 [23] and ImageNet [47]. The details of training dynamics of models with atoms from different time point during the training are shown in Figure 11 and Figure 12. Moreover, we examine the similarity between the two participated models shared the same initialization trained only with atoms on two different tasks. The results is shown in Figure 13 and Figure 14. The difference is less on the first few layers, but more on the middle layers. It reflects the middle layer is more critical than other layers, which is aligned with previous work [39].

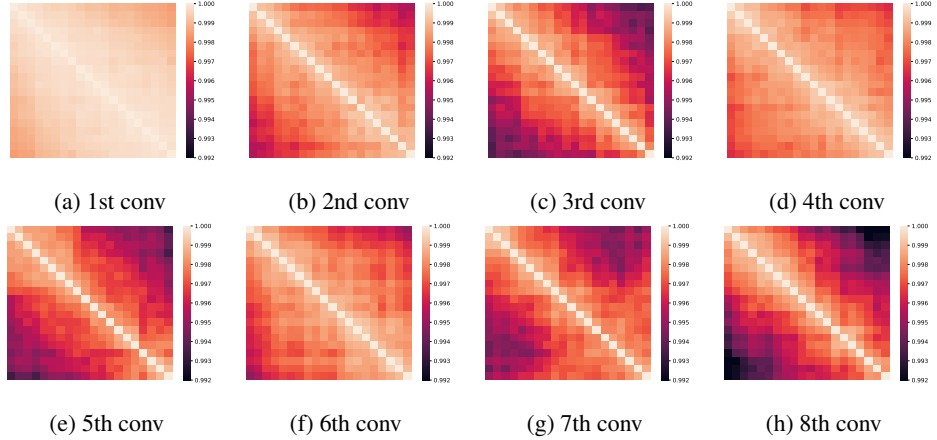

(a) 1st conv      (b) 2nd conv      (c) 3rd conv      (d) 4th conv

(e) 5th conv      (f) 6th conv      (g) 7th conv      (h) 8th conv

Figure 12: Similarity of VGG with atoms from different time point during the training.

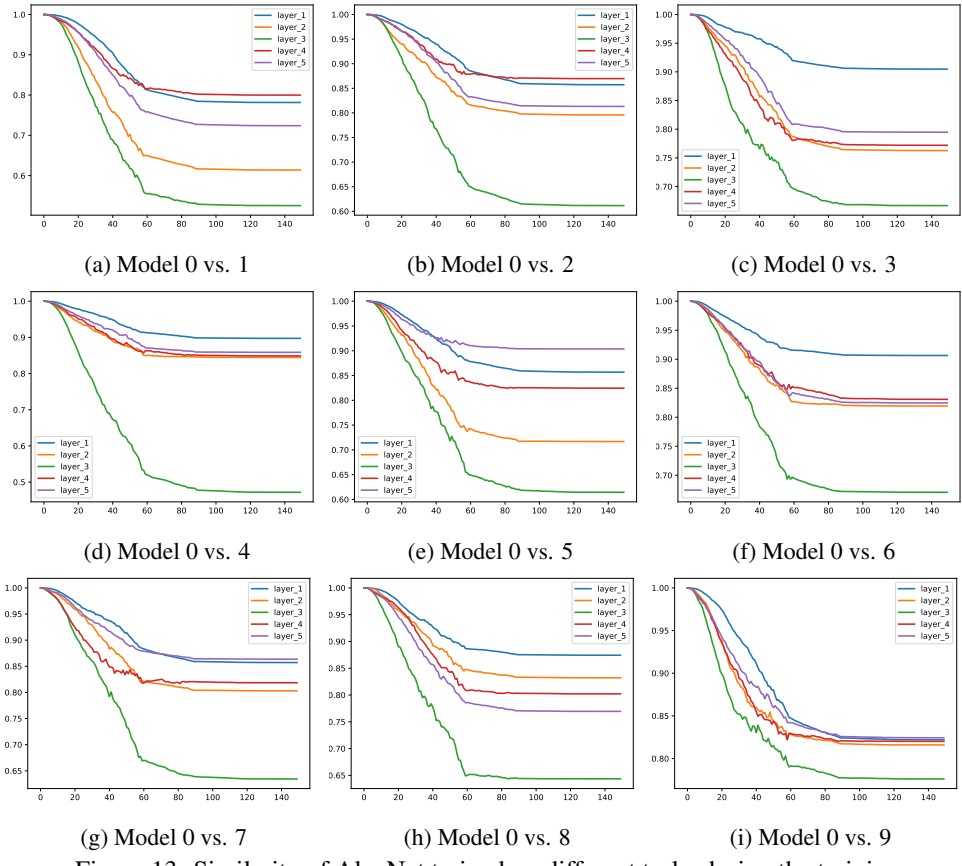

(a) Model 0 vs. 1      (b) Model 0 vs. 2      (c) Model 0 vs. 3

(d) Model 0 vs. 4      (e) Model 0 vs. 5      (f) Model 0 vs. 6

(g) Model 0 vs. 7      (h) Model 0 vs. 8      (i) Model 0 vs. 9

Figure 13: Similarity of AlexNet trained on different tasks during the training.

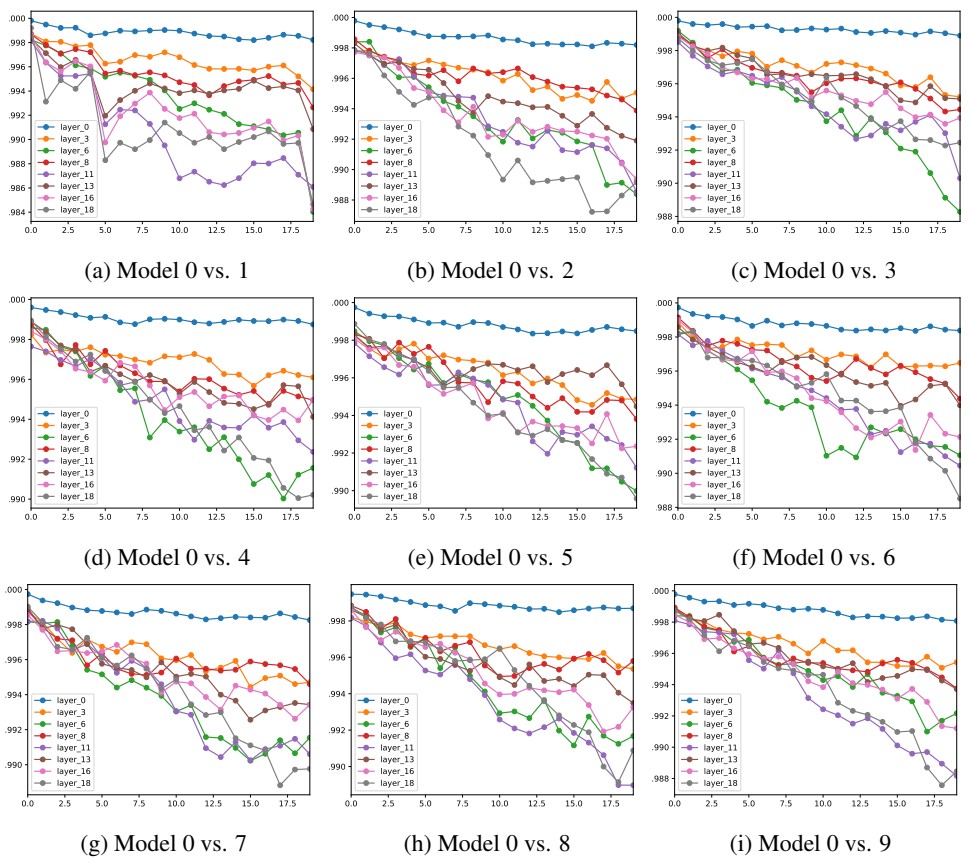

Figure 14: Similarity of VGG trained on different tasks during the training.

