540 $\qquad\qquad\qquad\qquad\qquad\qquad\qquad\qquad\qquad\qquad\qquad\qquad\qquad\qquad\qquad\qquad$ □

541 **Lemma A.5.** *For two matrices* $\mathbf{A}$, $\mathbf{B}$, *their frobenius norm satisfies,*

$$
\|\mathbf{AB}\|_F = \|\mathbf{A}\|_F \|\mathbf{B}\|_F \sqrt{1 - \frac{\Delta_1}{\|\mathbf{A}\|_F^2 \|\mathbf{B}\|_F^2}},
\tag{36}
$$

542 *where* $\Delta_1 = \sum_{ij}(\sum_k A_{ik}^2)(\sum_k B_{kj}^2) \cdot \sin^2(\langle A_{i:}, B_{:j}\rangle)$.

543 *Proof.* According to the definition of frobenius norm $\|\mathbf{A}\|_F = \sqrt{\sum_{ij}|A_{ij}|^2}$ we have,

$$
\|\mathbf{AB}\|_F = \sqrt{\sum_{ij}(\sum_k A_{ik}B_{kj})^2}.
\tag{37}
$$

544 Note that $(\sum_i x_i y_i)^2 = (\sum_i x_i^2)(\sum_i y_i^2) \cdot \cos^2(\langle x, y\rangle) = (\sum_i x_i^2)(\sum_i y_i^2) - (\sum_i x_i^2)(\sum_i y_i^2) \cdot$
545 $\sin^2(\langle x, y\rangle)$, where $\langle x, y\rangle$ is the angle of two vectors $x$ and $y$. We have,