# OpenReview forum: "Inner Product-based Neural Network Similarity"
_NeurIPS.cc/2023/Conference — NeurIPS 2023 poster_

### Official Review · Reviewer_qnN8 · 2023-06-15

**Soundness:** 3 good
**Presentation:** 3 good
**Contribution:** 3 good
**Rating:** 6
**Confidence:** 3

**Summary:**

This paper studies neural network similarity which is an important problem in many areas such as federated learning and continue learning. In this paper, the authors develop a new method to reduce NN representational similarity to filter subspace distance. Moreover, they present the effectiveness and efficiency of their algorithms in theory and practice.

**Strengths:**

1. The problem studied in this paper is fundamental.
2. The algorithm mentioned in this paper is simple and effective.
3. The experimental results show that the method developed in this paper is effective.
4. The authors present the effectiveness of algorithm in theory and practice.

**Weaknesses:**

1. The solution mentioned in this paper can be used for only CNNs.
2. The experimental section can be improved.

**Questions:**

1. Is it possible to extend this algorithm to other models? I am interested in why this method can be extended to transformer-based model.
2. As for the experimental section, is it possible to add experimental results over CIFAR-10, SVHN, and ImageNet?

**Limitations:**

If the authors can solve the issues mentioned in the Questions, I am willing to improve the rate.

---

> ### Author Rebuttal · Authors · 2023-08-08
>
> Thank you for your supportive comments.
>
> **Q1:** Is it possible to extend this algorithm to other models, such as transformer-based model?
>
> **A:** We can extend our method to other models, including transformer-based models. The proposed method relies on the decomposition of weight matrices, and sharing components with a substantial number of parameters among models while finetuning remaining parameter-efficient elements. This approach can thus be effectively applied to, for example, linear layers, major components of transformer-based models (Feedforward, Query, Key, and Value matrices). Specifically, by utilizing our proposed method in Section 2.2, we can decompose the weight of the linear layer $W \in \mathbb{R}^{c_{out} \times c_{in}}$ into two components: $W = \text{reshape}(\alpha \times D)$, where the atom coefficients are represented by $ \alpha \in \mathbb{R}^{c'\_{out} \times c'\_{in} \times m} $, and the atoms by $D \in \mathbb{R}^{m \times k \times k}$, and $c_{out} = c'\_{out} \times k$ and $c_{in} = c'\_{in} \times k$. For instance, assume $k=4, m=9$, weight matrix $w \in \mathbb{R}^{256 \times 64}$ is decomposed into $\alpha \in \mathbb{R}^{64\times 16\times 9}$ and $D\in \mathbb{R}^{9\times 4\times 4}$. Moreover, we finetune $D$ while fixing $\alpha$ when finetuning transformer models. Following the experimental setup in Section 3.4, we apply our method to continual learning with transformer-based models. The corresponding results are presented in the table below.
>
> |         | CIFAR100 | MFLOPs | Time (s) | GPU Memory (MB) |
> |---------|:------:|---------|:------:|---------|
> | ViT (base) |  75.17 $\pm$ 0.21  | - | - | - |
> | +CCA     |  77.28 $\pm$ 0.09 |  4.13 $\times 10^7$  |  46.84  |  1181  |
> | +CKA     |  76.67 $\pm$  0.13|  1.81 $\times 10^5$  |  35.08  |  1209  |
> | +Ours     |  **78.16  $\pm$ 0.05** | **0.015**   |  **0.35**  |  **0**  |
>
>
> Our method also exhibits a strong correlation with CCA/CKA in transformer-based models. The correlation between our method and CCA/CKA is shown in the table below, and Figure 1 (a)(b) in the attached PFD of the general response. Our experiment is built on top of the code [1].
>
> |         | Correlation |
> |---------|:------:|
> | CCA     |  0.9443  |
> | CKA     |  0.9079  |
>
> Moreover, by adopting the experimental setting outlined in Section 3.3, our method enables the measurement of task similarity using transformer-based models. In this specific experiment, we employed 100 models, and the CIFAR-100 dataset was divided into 20 subtasks, with each subtask containing 5 classes. Each subtask was shared by 5 models. As demonstrated in Figure 1 (c) in the attached PDF of the general response, every group of 5 models that share the same task shows a notably high similarity among themselves. The results indicate that our method can effectively measure the similarities of transformer-based models.
>
> On the other hand, since the above decomposition equation remains the same form as the convolution considered in the paper, our theoretical results can also be extended to transformer models with linear layers. However, a rigorous application of our method to transformer-based models would require further study, which we leave for further study.
>
>
> **Q2:** Is it possible to add experimental results apart from CIFAR, SVHN, and ImageNet?
>
> **A:** We evaluate the performance of our method on three different datasets: CelebA[2], Oxford Flower[3], and Food-101[4]. The experimental setup remains consistent with the one described in Section 3.1 of the paper. As shown in the table below, our method consistently shows high correlations with CCA and CKA on three datasets.
>
> |         | CelebA [2] | Flower [3] | Food [4] |
> |---------|:------:|--------|:----:|
> | CCA     |  0.9014  | 0.9155 | 0.9901 |
> | CKA     |  0.8766  | 0.8831 | 0.9266 |
>
> Reference:
> 1. https://github.com/kentaroy47/vision-transformers-cifar10
> 2. Liu, Ziwei and Luo, Ping and Wang, Xiaogang and Tang, Xiaoou. "Deep Learning Face Attributes in the Wild." Proceedings of International Conference on Computer Vision (ICCV), 2015.
> 3. Nilsback, M-E. and Zisserman, A. "Automated flower classification over a large number of classes." Proceedings of the Indian Conference on Computer Vision, Graphics and Image Processing, 2008.
> 4. Bossard, Lukas and Guillaumin, Matthieu and Van Gool, Luc. "Food-101 -- Mining Discriminative Components with Random Forests." European Conference on Computer Vision, 2014.

---

### Official Review · Reviewer_PBCY · 2023-07-08

**Soundness:** 3 good
**Presentation:** 4 excellent
**Contribution:** 3 good
**Rating:** 6
**Confidence:** 3

**Summary:**

In their paper, the authors introduce a novel approach to significantly decrease the computational cost of representational similarity analysis in CNNs by transitioning it to filter subspace distance evaluation. Their proposed filter subspace-based similarity is both theoretically and empirically demonstrated to display a robust linear correlation with prevalent probing-based metrics.

**Strengths:**

1.	The approach offers significant improvements in efficiency and robustness.
2.	The paper is very clear, especially the methodology part is clearly written and easy to understand.
3.	The proposed method is simple, making it easy to implement.


**Weaknesses:**

1.	Could you elaborate on how Assumption 2.6 is applicable to complex real-world datasets? It appears to be a strong assumption. For instance, consider a simple matrix $$\begin{pmatrix}0 & 0\\\ 1 & 0.9\end{pmatrix}$$, the assumption does not hold true.
2.	No obvious weakness in my mind now.


**Questions:**

1.	On line 115, is m a hyperparameter? If yes, is the model performance sensitive to the selection of m?
2.	In Figure 2b, it appears that the points located in the center are less correlated (or more distant from the diagonal blue line) than the points positioned at the ends. Could you explain why this might be?
3.	With regard to Proposition 2.1, would the inequality become significantly looser if many extreme values exist in X?
4.	On line 148, the proposed method suggests averaging layer-wise similarities for network-wise similarity. Is it possible that some layers might be more significant than others? For instance, should the final few layers carry more importance?
5.	The paper [1] outlines certain limitations of CCA/CKA in Section 3, for example, randomly permuting the order of pixels (either at the input or in the latents) is an orthogonal transform, and thus does not affect the CKA. However, this destroys the spatial structure of the input, and intuitively should affect the “representation quality.”, does this apply to the proposed method?

[1]Bansal, Yamini, Preetum Nakkiran, and Boaz Barak. "Revisiting model stitching to compare neural representations." Advances in neural information processing systems 34 (2021): 225-236.


**Limitations:**

Authors do not discuss the limitation of their work. Based on the information presented, it does not appear that this work has any negative societal impacts.

---

> ### Author Rebuttal · Authors · 2023-08-08
>
> Thank you for your supportive comments.
>
> **W1:** Could you elaborate on how Assumption 2.6 is applicable to complex real-world datasets? Consider a simple matrix $\begin{pmatrix} 0 & 0 \\\\ 1 & 0.9 \end{pmatrix}$, the assumption does not hold true.
>
> **A:** Our assumption states that different channels in the features are less correlated. In specific, every diagonal component in $Z_u^T Z_u \in \mathbb{R}^{c\times c}$ represents the variances within each channel, and an off-diagonal element indicate the correlation between two channels. Therefore, the 2x2 matrix shown by the reviewer may not be valid in practice, as there is likely no 0 in the diagonal elements, and little chance to have two channels completely correlated.
>
> In complex datasets, it may be hard to have features that satisfy the assumption directly. As shown in the Appendix line 628-631, we further propose a regularization term to reduce channel-wise correlation, which improves the correlation with CCA in Figure 8(b) and matches our theoretical findings.
>
>
>
> **Q1:** On line 115, is m a hyperparameter? If yes, is the model performance sensitive to the selection of m?
>
> **A:** Yes, $m$ is a hyperparameter. In our paper, we choose $m=9$. The similarity of models remains robust across different choices of $m$. In this experiment, we follow the setting in Section 3.1. Specifically, we train 10 models separately on 10 sub-tasks of the CIFAR-100 dataset. In the table below, we present the correlation between CCA and the filter subspace similarity with different choices of $m$, the standard deviation of the correlations is only 0.0027.
>
> |   m   | 3 | 6 | 9 | 12 |
> |---------|:------:|---------|:------:|---------|
> | correlation w/ CCA    |  0.9313  | 0.9289  | 0.9327  | 0.9366  |
>
>
>
> **Q2:** In Figure 2b, it appears that the points located in the center are less correlated (or more distant from the diagonal blue line) than the points positioned at the ends. Could you explain why this might be?
>
> **A:** We hypothesize that when networks are more intrinsically dissimilar, the distance between atoms $D_u, D_v$ becomes larger, i.e., $\text{cos}(D_u, D_v)$ becomes smaller. In this case, the linear correlation between our method and CCA would be slightly affected, as stated in line 195. Yet, we still show a reasonably strong correlation with CCA in both high-similarity and low-similarity cases. which is shown in the following table as the similarity is in the range [0.4, 0.5].
>
> |         | Correlation |
> |---------|:------:|
> | CCA     |  0.9231  |
> | CKA     |  0.9178  |
>
>
>
> **Q3:** With regard to Proposition 2.1, would the inequality become significantly looser if many extreme values exist in X?
>
> **A:** As we only consider real image datasets, the values in $X$ are normalized and bounded. Thus it is unlikely to have extreme values in it.
>
>
>
> **Q4:** On line 148, the proposed method suggests averaging layer-wise similarities for network-wise similarity. Is it possible that some layers might be more significant than others? For instance, should the final few layers carry more importance?
>
> **A:** Since different layers can carry different information, we agree that there would be a more effective weighted scheme to aggregate layer-wise similarities. Since it is likely non-trivial to decide weights for different layers, we leave it for future study.
>
>
>
> **Q5:** The paper [1] outlines certain limitations of CCA/CKA in Section 3, for example, randomly permuting the order of pixels (either at the input or in the latents) is an orthogonal transform, and thus does not affect the CKA. However, this destroys the spatial structure of the input, and intuitively should affect the “representation quality.”, does this apply to the proposed method?
>
> **A:** Our method is immune to those "attacks" on CCA/CKA as we don't rely on probing data to compute feature similarity. Our method focuses on the intrinsic similarity of models with inner-product of decomposed model parameters. It is agnostic to distortion on probing data. We show similar experiments as [1] in Figure 3(b), where CCA and CKA are severely affected by the choice of the probing data and our atom-based similarity remains robust.

---

> > ### Comment · Reviewer_PBCY · 2023-08-16
> >
> > The authors addressed most of my concerns. I would keep the score as "Weak Accept" but increase the confidence to 3.

---

> > > ### Author Response · Authors · 2023-08-17
> > >
> > > Thanks for the positive feedback and acknowledgment. We sincerely appreciate your time and efforts.

---

### Official Review · Reviewer_naiq · 2023-07-09

**Soundness:** 4 excellent
**Presentation:** 4 excellent
**Contribution:** 3 good
**Rating:** 7
**Confidence:** 4

**Summary:**

This paper presents a new approach for measuring the similarity between neural network models based on a new efficient metric defined layer-wise on filter atoms.
In particular, the metric is defined as the cosine similarity between this filter atoms that can be implemented as the normalised inner product of the filter atom vectors.
It is shown that 1) the probing-based similarity CCA is upper-bounded by the proposed metric up to a scaling factor, where both CCA and the factor are data dependant, and 2) under the assumption that the features in a layer have low correlation, there is an approximately linear relation between CCA and the proposed metric.
Experimental results show that the proposed approach can be effectively used to study the learning dynamics of deep neural networks. The method has been further applied to Personalized Federated Learning and Continual Learning and improves the results over CCA and CKA similarity metrics.


**Strengths:**

- An original theoretically-grounded approach for measuring layer-wise similarity between two neural network models
- Evaluation on different applications and good results

**Weaknesses:**

- Comparison only with CCA and CKA-based similarity metrics
- Assumption 2.6. is quite strong. Although your empirical results seem to corroborate this assumption, in practice I believe there can be strong correlations between different features within a layer.
- Only AlexNet and VGG are used (a part from the experiment showing the correlation between Grassmann similarity and the proposed filter subspace similarity). It would have been interesting, if the proposed method also performs well for other models, e.g. ResNet models.






**Questions:**

- In the experiments, the overall similarity between two neural networks is computed as the average over the layer-wise similarities. However, intuitively not all layers are equally important for a given task. Would it make sense to use different weights for different layers (i.e. a weighted average)? Would that change the overall results?


**Limitations:**

See weaknesses.

---

> ### Author Rebuttal · Authors · 2023-08-08
>
> Thank you for your supportive comments.
>
> **W1:** Comparison only with CCA and CKA-based similarity metrics
>
> **A:** We want to claim that CCA and CKA are two representative feature-based network similarity measures that are adopted in many works [1,2,3]. Showing a strong linear correlation with CCA and CKA is significant for demonstrating the effectiveness of our method.
>
> Reference:
> 1. Raghu, Maithra, et al. "Do vision transformers see like convolutional neural networks?." Advances in Neural Information Processing Systems, (2021).
> 2. Neyshabur, Behnam, Hanie Sedghi, and Chiyuan Zhang. "What is being transferred in transfer learning?." Advances in neural information processing systems, (2020).
> 3. Yang, Xingyi, et al. "Deep model reassembly." Advances in neural information processing systems, (2022).
>
> **W2:** Assumption 2.6. is quite strong. Although your empirical results seem to corroborate this assumption, in practice I believe there can be strong correlations between different features within a layer.
>
> **A:** As shown in the Appendix line 628-631, we further design a regularization loss to reduce the correlation between features, which improves the correlation between our method and CCA in Figure 8(b) and matches our theoretical findings.
>
> **W3:** Only AlexNet and VGG are used (apart from the experiment showing the correlation between Grassmann similarity and the proposed filter subspace similarity). It would have been interesting, if the proposed method also performs well for other models, e.g. ResNet models.
>
> **A:** As shown in Figure 2(b) and Figure 2 (Table), we show strong correlations between our method and CCA/ CKA with ResNet-18. Besides, we also show the effectiveness of our method with ViT in our general response **GR1**.
>
> Following the experimental setup in Section 3.4, we apply our method to continual learning with transformer-based models. The corresponding results are presented in the table below.
>
> |         | CIFAR100 | MFLOPs | Time (s) | GPU Memory (MB) |
> |---------|:------:|---------|:------:|---------|
> | ViT (base) |  75.17 $\pm$ 0.21  | - | - | - |
> | +CCA     |  77.28 $\pm$ 0.09 |  4.13 $\times 10^7$  |  46.84  |  1181  |
> | +CKA     |  76.67 $\pm$  0.13|  1.81 $\times 10^5$  |  35.08  |  1209  |
> | +Ours     |  **78.16  $\pm$ 0.05** | **0.015**   |  **0.35**  |  **0**  |
>
>
> **Q1:** In the experiments, the overall similarity between two neural networks is computed as the average over the layer-wise similarities. However, intuitively not all layers are equally important for a given task. Would it make sense to use different weights for different layers (i.e. a weighted average)? Would that change the overall results?
>
> **A:** We agree that there would be a more effective weighted scheme to aggregate layer-wise similarities. Since it is likely non-trivial to decide weights for different layers, we leave it for future study.

---

### Official Review · Reviewer_8het · 2023-07-25

**Soundness:** 3 good
**Presentation:** 3 good
**Contribution:** 2 fair
**Rating:** 5
**Confidence:** 4

**Summary:**

This paper proposes an approach for evaluating neural network similarity by decomposing convolution layer into linear combination of atom filters. The basic idea derives from [33] and the paper extend the approach for continual learning (CL), to federated learning (FL). The paper provides both empirical and theorical evidence and shows the time-efficiency of the proposed method. However, the application scenarios are relatively limited.

**Strengths:**

S1. The paper extends [33], which is originally oriented for continual learning (CL), to federated learning (FL), and compares with CCA and CKA.

S2. In limited cases, the proposed method is more efficient than feature-based similarity evaluation (e.g., CCA [40] and CKA [19])

S3. In limited cases, the proposed method can achieve more accurate similarity evaluation and can provide theoretical justifications.


**Weaknesses:**

W1. The proposed method has strong restrictions in limited cases.
W1.1 The proposed method is restricted to only convolutional layers, and it further needs different neural networks using the same atom coefficients. That means the proposed method is hard to be used in the similarity evaluation of common neural networks.

W1.2 The proposed method is restricted to cases where the model architecture must be identical and the model parameters much be similar. For example, in Fig. 2 (Correlation between filter subspace similarity and other approaches), all the three methods achieve a high estimated similarity up to 0.9. What about the cases of lower similarity? The author should present more results for the lower estimated similarity. Thus, it’s unclear whether the proposed method is as flexible as the baselines CCA and CKA, which can be used to compare neural networks with different depths or trained by different datasets. Whether the proposed method can only be used in FL and CL, where the models as well as the parameters are inherently similar?

W1.3 The theoretical evidence in this paper is limited to single-layer convolution, particularly, without nonlinearity. Proposition 2.4 (line 152) is restricted to orthogonal matrices and the case of k^2 =m.

W2 In FL/CL, the proposed method is time-efficient, but that time is not a severe bottleneck in FL/CL. Furthermore, by nature, the proposed method vectorizes the model parameters and computes the similarities of vectors, which are widely used in multi-task learning.

W3. The proposed method has already been proposed in [33].


**Questions:**

Q1. In the experiment, CKA and CCA compute which layer computation? How does the proposed method consider merging the result of different layers? By concatenation or summation? This paper only discusses the case of single-layer convolution.

Q2. What’s the essential difference between the proposed method and directly computing the cosine similarity and inner product of parameters?

---

> ### Author Rebuttal · Authors · 2023-08-08
>
> Thanks for your constructive comments. We address all your concerns in the following.
>
> **W1.1** The proposed method is restricted to only convolutional layers, and it needs different NNs using the same atom coefficients.
>
> **A:** Although our method focuses on convolutional filter subspace, it can also be easily extended to other types of layers, e.g., linear layers. As explained in our general response **GR1**, our approach can be effectively extended to **transformer-based models** in continual learning experiments while still maintaining a strong correlation with CCA/CKA.
>
> We also would like to state that fixing a large number of parameters of a pretrained model while finetuning a small portion is a common paradigm in the field of modeling visual knowledge, e.g., images. The shared parameters $A_i$ can be viewed as the fixed part of a model pretrained on a significant amount of data, and the $B_i$ is a set of a small number of parameters tuned to fit downstream tasks. In other words, we can potentially obtain pretrained atom coefficients on large datasets and finetuning multiple models on downstream tasks for better performances. Therefore, we argue that, although a new filter subspace view to NN is adopted in the paper, sharing atom coefficients across models does not deviate from the above common paradigm, and poses no major limitation.
>
> **W1.2** The author should present more results for the lower estimated similarity.
> Whether the proposed method can only be used in FL and CL, where the models as well as the parameters are inherently similar?
>
> **A:** Our method also applies to lower-similarity cases. We conduct an additional experiment to show the correlation between our method and CCA/CKA in low-similarity cases. In our experiment, we train neural networks on two distinct datasets, Oxford Flower and Food-101. For each dataset, we trained 10 models and computed the similarity between models trained on different datasets. The results show a low similarity range of [0.4, 0.5]. In this case, our similarity measure still exhibits a consistently high correlation with CCA/CKA, as shown in Figure 2 in the attached PDF of our general response.
>
> |         | Correlation |
> |---------|:------:|
> | CCA     |  0.9231  |
> | CKA     |  0.9178  |
>
> When considering the network architecture, it is important to highlight the increasing popularity of the pretraining-finetuning paradigm today, as discussed above, where the network structure remains consistent. In our framework, we have pretrained atom coefficients and fine-tuned the filter atoms with various architectures, demonstrating the versatility of our method, which can be applied not only to FL and CL but also to this commonly used setting.
>
> **W1.3** The theoretical evidence in this paper is limited to single-layer convolution without nonlinearity. Proposition 2.4 is restricted to orthogonal matrices and the case of k^2 =m.
>
> **A:** In Proposition 2.4, $D_u, D_v$ are not required to be square matrices. As shown in line 502 in the Appendix, by saying 'orthogonal matrices' we actually assume atoms matrices satisfy $D_u^T D_u=I$, i.e. their columns are orthonormal. We will revise it to 'Assume $D_u, D_v$ satisfy $D_u^T D_u=D_v^T D_v=I$' in Proposition 2.4 for clarification. Note that orthonormality makes equality happen in Proposition 2.4, a strong correlation can still be observed without the orthonormality requirement as shown in Figure 2 Table.
>
> We can extend our analysis to a non-linear layer with approximation to the non-linearity. Consider the non-linear layer, $Z=\sigma(\alpha X D)$, we can approximate it with two linear terms as $Z=\alpha X D + \alpha X' D=\alpha \tilde{X} D$, where $X'=\alpha^{+}(\sigma(\alpha X D) - \alpha X D) D^+$, $\alpha^{+}, D^+$ denote psuedoinverses of $\alpha, D$. For the derivation of Theorem 1 and 2, we can simply replace $X$ with $\tilde{X}$ and the rest remains the same. In this way, our main theoretical results still hold.
>
> Finally, we simplify our analysis with a single-layer setting, yet it can be easily extended to multi-layer.
>
> **W2** In FL/CL, the proposed method is time-efficient, but...
>
> **A:** It is infeasible to directly assess network similarities by vectorizing their weights and computing the inner product, since a permutation matrix is required for alignment [1]. Our method leverages the decomposition structure and can directly evaluate network similarities with filter atom inner-products.
>
> Our method is not restricted to FL/ CL tasks. We only adopt FL/ CL to demonstrate our method in multi-model scenarios. In a large multi-model system where there are frequent needs to compare similarities for, e.g., deciding whether to train a new model or reuse an old one given a new task, the proposed method can dramatically reduce the latency and costs of model comparisons.
>
> **W3** The proposed method has already been proposed in [33].
>
> **A:** [33] proposes to use the Grassmann distance of filter atoms to assess network similarity as a heuristic approach with **no theoretical justification**. In this work, we simplify the atom-based similarity to inner-product computation and provide both theoretical and empirical results to demonstrate its equivalence with popular feature-based similarities CCA/CKA.
>
> **Q1.** How does the proposed method consider merging the result of different layers?
>
> **A:** As we explained in line 147-148, we simply average layer-wise similarities for the network-wise similarity.
>
> **Q2.** What’s the essential difference between the proposed method and directly computing the cosine similarity and inner product of parameters?
>
> **A:** As explained in our general response **GR2**, there is a substantially low correlation between CCA/CKA and the weight inner-product, suggesting that directly using the inner product of parameters is unlikely to effectively measure model similarity.
>
> Reference:
> 1. Ainsworth, Samuel, et al. "Git Re-Basin: Merging Models modulo Permutation Symmetries." ICLR. 2022.

---

> > ### Comment · Reviewer_8het · 2023-08-17
> >
> > The reviewer thanks the clarification of the authors. As far as the my concern, the theoretical contributions are limited by nature, however, I improve the rating of this paper from 4 to 5, because of the extension to FL/CL.

---

> > > ### Author Response · Authors · 2023-08-17
> > >
> > > We sincerely thank reviewer 8het for the time and effort invested in evaluating our work.

---

### Author Rebuttal · Authors · 2023-08-08

### General Responses
*We provide figures for the additional required experimental results in the attached PDF.*

**GR1: Extend our algorithm to transformer-based models.**

Although our method focuses on convolutional filter subspace due to the highly compact size of resulting filter subspace elements (atoms), it can also be easily extended to other types of layers, e.g., linear layers. In essence, our findings rely on: 1) weights $W_i$ are decomposed into two components, $W_i=A_i\times B_i$; 2) in a group of models, one component $B_i$ is fine-tuned while the other component $A_i$ remains fixed. For example, one can apply our method to transformer-based models with decomposed weight matrices in linear layers (Feedforward, Query, Key, and Value matrices).
Specifically, by utilizing our proposed method in Section 2.2, we can decompose the weight of the linear layer $W \in \mathbb{R}^{c_{out} \times c_{in}}$ into two components: $W = \text{reshape}(\alpha \times D)$, where the atom coefficients are represented by $\alpha \in \mathbb{R}^{c'\_{out} \times c'\_{in} \times m}$, and the atoms by $D \in \mathbb{R}^{m \times k \times k}$, and $c_{out} = c'\_{out} \times k$ and $c_{in} = c'\_{in} \times k$. For instance, assume $k=4, m=9$, weight matrix $w \in \mathbb{R}^{256 \times 64}$ is decomposed into $\alpha \in \mathbb{R}^{64\times 16\times 9}$ and $D\in \mathbb{R}^{9\times 4\times 4}$. Moreover, we finetune $D$ while fixing $\alpha$ when finetuning transformer models.
Following the experimental setup in Section 3.4, we apply our method to continual learning with transformer-based models. As shown in the table below, our approach can be effectively extended to transformer-based models in continual learning experiments while still maintaining a strong correlation with CCA/CKA.

|         | CIFAR100 | MFLOPs | Time (s) | GPU Memory (MB) |
|---------|:------:|---------|:------:|---------|
| ViT (base) |  75.17 $\pm$ 0.21  | - | - | - |
| +CCA     |  77.28 $\pm$ 0.09 |  4.13 $\times 10^7$  |  46.84  |  1181  |
| +CKA     |  76.67 $\pm$  0.13|  1.81 $\times 10^5$  |  35.08  |  1209  |
| +Ours     |  **78.16  $\pm$ 0.05** | **0.015**   |  **0.35**  |  **0**  |

|         | Correlation |
|---------|:------:|
| CCA     |  0.9443  |
| CKA     |  0.9079  |

Moreover, by adopting the experimental setting outlined in Section 3.3, our method enables the measurement of task similarity using transformer-based models. In this specific experiment, we employed 100 models, and the CIFAR-100 dataset was divided into 20 subtasks, with each subtask containing 5 classes. Each subtask was shared by 5 models. As demonstrated in Figure 1 (c) in the attached PDF, every group of 5 models that share the same task shows a notably high similarity among themselves. The results indicate that our method can effectively measure the similarities of transformer-based models.

On the other hand, since the above decomposition equation remains in the same form as the convolution considered in the paper, our theoretical results can also be extended to transformer models with linear layers. However, a rigorous application of our method to transformer-based models would require further study, which we leave for further study.

**GR2: The fundamental difference between our method and directly computing the cosine similarity or inner product of weight parameters.**

It is infeasible to directly assess network similarities by computing cosine similarity and inner product of parameters, since a permutation matrix is required for alignment [1]. Our method leverages the decomposition structure and can directly evaluate network similarities with filter atom inner-products.

Here we present an ablation study to show the correlation between CCA/ CKA with both the weight inner-product and our atom inner product. We adopt the same settings in Section 3.1. The table presented below demonstrates that there is a substantially low correlation between CCA/CKA and the weight inner-product, suggesting that directly using the inner product of parameters is unlikely to effectively measure model similarity. The results are also presented in Figure 3 in the attached PDF.

|         | Inner-product of weights | Inner-product of atoms |
|---------|:------:|:------:|
| Correlation w/ CCA     |  0.0803  | **0.9327** |
| Correlation w/ CKA     |  0.2924  | **0.9550** |

Reference:
1. Ainsworth, Samuel, et al. "Git Re-Basin: Merging Models modulo Permutation Symmetries." International Conference on Learning Representations. 2022.

---

### Decision · Program_Chairs · 2023-09-21

**Decision:**

Accept (poster)

**Comment:**

This paper proposes a novel method for computing neural network similarity efficiently, which decomposes each filter into a linear combination of filter atoms, and then treats the coefficient similarity as the network similarity. The paper provides theory and also empirical results to verify the efficiency and effectiveness, compared with previous methods, such as CCA.